# Multi-class rice seed recognition based on deep space and channel residual network combined with double attention mechanism

Tingyuan Zhang[1], Changsheng Zhang ●[1]*, Zhongyi Yang[2], Meng Wang[1], Fujie Zhang[3], Dekai Li[1], Sen Yang[4]

**1** Faculty of Information Engineering and Automation, Kunming University of Science and Technology, Kunming, Yunnan, China, **2** Biotechnology and Germplasm Resources Institute, Yunnan Academy of Agricultural Sciences, Kunming, Yunnan, China, **3** Faculty of Modern Agriculture Engineering, Kunming University of Science and Technology, Kunming, Yunnan, China, **4** Department of Computer Science and Technology, West Yunnan University of Applied Sciences, Dali, Yunnan, China

\* 13674726610@163.com

## Abstract

Accurately recognizing rice seed varieties poses significant challenges due to their diverse morphological characteristics and complex classification requirements. Traditional image recognition methods often struggle with both accuracy and efficiency in this context. To address these limitations, this study proposes the Deep Space and Channel Residual Network with Double Attention Mechanism (RSCD-Net) to enhance the recognition accuracy of 36 rice seed varieties. The core innovation of RSCD-Net is the introduction of the Space and Channel Feature Extraction Residual Block (SCR-Block), which improves inter-class differentiation while minimizing redundant features, thereby optimizing computational efficiency. The RSCD-Net architecture consists of 16 layers of SCR-Blocks, structured into four convolutional stages with 3, 4, 6, and 3 units, respectively. Additionally, a Double Attention Mechanism (A$^2$Net) is incorporated to enhance the network's global receptive field, improving its capacity to distinguish subtle variations among seed types. Experimental results on a self-collected dataset demonstrate that RSCD-Net achieves an average accuracy of 81.94%, surpassing the baseline model by 4.16%. Compared with state-of-the-art models such as InceptionResNetV2, ConvNeXt, MobileNetV3, and Swin Transformer, RSCD Net has improved by 1.17%, 3%, 24.72%, and 13.22%, respectively, showcasing its superior performance. These findings confirm that RSCD-Net provides an effective and efficient solution for rice seed classification, offering a promising reference for addressing similar fine-grained recognition challenges in agricultural applications.

**Data availability statement:** All relevant data are within the paper and its Supporting Information files.

**Funding:** This thesis supported by National Natural Science Foundation of China (62062048) and China Yunnan Province Science and Technology Plan Project(202201AT070113), The fund sponsor plays the role of providing financial support in this article, and is the fourth author of this article, Meng WANG.

**Competing interests:** The authors have declared that no competing interests exist.

## Introduction

Rice is a crop of immense economic and nutritional significance, cultivated and consumed extensively around the globe. It is typically available in both fresh and processed forms, playing a crucial role in boosting farmers' incomes and enhancing agricultural systems. Rich in essential nutrients such as carbohydrates, proteins, fats, and B vitamins, rice is vital for maintaining human health. Asia is the primary region for rice consumption, where it serves as a staple food, particularly in populous countries like India and China. Additionally, several countries in Africa, Latin America, and the Caribbean also incorporate rice into their diets. Over half of the countries worldwide regard rice as their staple food, underscoring its critical role in the global food supply [1]. In addition to being a food source, rice is significant for feed production, agricultural research, industrial raw materials, and the bioenergy sector. To enhance the quality and nutritional value of rice, researchers have developed thousands of varieties, including indica, japonica, and giant rice. Since the specific variety can greatly influence the quality and nutritional content, accurate identification is essential for seed producers, breeders, and consumers alike. Moreover, rice can be processed into various by-products, such as yellow wine and oils that are rich in unsaturated fatty acids. Given that the quality of rice seeds directly affects both yield and quality [2], precise identification and differentiation of these seeds are particularly important.

Traditional classification methods, such as manual sorting and biomolecular classification, are effective to a degree but encounter several challenges, including high costs, low efficiency, and the potential for seed damage. To overcome these issues, research has increasingly shifted toward machine vision and deep learning image recognition technologies. These advanced techniques significantly enhance both the efficiency and accuracy of seed classification by analyzing various characteristics, such as shape, color, chaff, size, and texture. Common classifiers employed for this purpose include Support Vector Machines (SVM), k-Nearest Neighbors, Artificial Neural Networks (ANN), Deep Neural Networks (DNN) [3], and Multi-column Back Propagation Neural Networks (BPNN) [4]. For example, Ruslan developed a method utilizing RGB image analysis and machine learning to successfully identify weedy rice among conventional rice seeds [5]. Similarly, Zhou and colleagues integrated Hyperspectral Imaging (HSI) with various machine learning techniques, including Support Vector Machines, Random Forests, and Gradient Boosting Classifiers, to anticipate beet seed germination [6]. While traditional image processing methods perform well in a variety of contexts, they often face limitations in classification accuracy and efficiency, especially when dealing with complex edge feature recognition tasks.

The integration of deep learning techniques has markedly advanced the field of image processing, particularly in crop classification. These techniques not only offer efficient and robust solutions but also excel at image recognition tasks, positioning them as a key focus in seed variety classification research. For example, the RiceNet system developed by Din N M U et al [7]. enhanced the accuracy of identifying five distinct rice varieties, achieving a prediction accuracy of 94%. This performance exceeds that of traditional machine learning methods, such as HOG-SVM and

SIFT-SVM, as well as pre-trained deep learning models like InceptionV3 and ResNetInceptionV2. Koklu M et al. analyzed 75,000 grain images using Artificial Neural Networks (ANN), Deep Neural Networks (DNN), and Convolutional Neural Networks (CNN) [8]. Their work resulted in a remarkable classification success rate of up to 100%, with CNN algorithms demonstrating particularly strong performance. Sharma A et al. introduced the iRSVPred model, which focuses on identifying ten major basmati rice varieties in India [9]. This model achieved perfect accuracy on both the training and internal validation sets, while recognition rates on external validation datasets reached or exceeded 80%. Lin P et al. compared traditional methods with Deep Convolutional Neural Networks (DCNN), finding that DCNN significantly improved the accuracy of rice grain image classification to 95.5% [10]. Additionally, Panmuang M et al. employed the VGG16 model for image screening of five rice varieties, attaining an accuracy of 85% [11]. Further research by Kiratiratanapruk K et al. indicated that utilizing the InceptionResNetV2 deep learning model for classifying 14 rice varieties resulted in an accuracy of 95.15%, outpacing traditional statistical learning methods [12]. Finally, Gilani G et al. developed an 18-layer convolutional neural network specifically for classifying seven major rice varieties cultivated in Pakistan, achieving 100% classification accuracy for each seed type [13].

Three-dimensional information provides a richer and more accurate representation of appearance features compared to two-dimensional images. Qian et al. utilized three-dimensional point cloud data alongside deep learning networks to classify eight varieties of rice, achieving an average accuracy of 89.4%, which marks a 1.2% improvement over the PointNet model [14]. Meanwhile, He et al. developed a multimodal fusion detection method based on an enhanced voting approach. This method combines two-dimensional and three-dimensional prediction probabilities through weighted averaging, resulting in an impressive average accuracy of 97.4% [15].

High-spectral imaging systems offer both spatial and spectral information, significantly enhancing classification accuracy. Chatnuntawech I et al. integrated hyperspectral imaging with deep convolutional neural networks (DCNN) for rice seed classification, achieving an average accuracy of 91.09% [16]. Tang et al. developed a hyperspectral image classification model based on a multilayer perceptron (MLP) network and residual learning, successfully distinguishing three types of processed rice seeds with an impressive accuracy of 98.48% [17]. Jin et al. employed high-spectral techniques alongside deep learning models such as LeNet, GoogleNet, and ResNet to classify ten varieties of rice seeds, with the ResNet model attaining the highest accuracy at 86.08% [18].

Research on the application of deep learning in rice seed classification and recognition remains limited, with few studies examining more than 15 rice varieties. In our study, we developed a dataset that encompasses 36 distinct rice seed varieties. Because rice classification belongs to fine-grained classification [19], and the characteristics between these varieties are often very similar, there may also be significant differences within the same variety, which adds complexity to the classification and recognition process. The outcomes from experiments using deep learning networks, such as ResNet, ConvNext, and InceptionResNetV2, as referenced in existing rice classification studies, have proven unsatisfactory when applied to our dataset. Additionally, while research that integrates hyperspectral imaging and 3D point cloud data may yield improved results, it also incurs higher costs. Moreover, this paper addresses another crucial challenge: many current studies depend on large quantities of training samples to train their networks. In contrast, our study seeks to effectively classify 36 different rice seed varieties using a more limited training set of only 240 samples per variety. Employing deep learning for the classification of crop seeds presents an efficient and straightforward approach to seed variety classification tasks. However, the exploration of deep learning applications specifically for rice variety classification remains underdeveloped. This study seeks to enhance the recognition accuracy of 36 rice seed varieties by leveraging deep learning technology and introducing a model titled the Deep Space and Channel Residual Network Combined with Double Attention Mechanism (RSCD-Net). The study commences with the design of a Space and channel Feature Extraction Residual Block (SCR-Block), which aims to amplify inter-class distinctions while minimizing redundancy in feature information, thereby streamlining model computation. The architecture comprises 16 residual modules, supplemented by batch normalization (BN) layers, as well as both max and average pooling layers. These residual modules are organized

into a comprehensive convolutional structure featuring configurations of 3, 4, 6, and 3 to optimize feature extraction. By integrating a double attention mechanism recognized as Double Attention Networks (A²Net), the model's global receptive field is significantly enhanced, leading to improved recognition capabilities for various seed varieties. In conclusion, this study proposes a deep learning-based method for rice seed classification. This innovative approach addresses the challenges commonly associated with traditional classification methods, such as high costs, time consumption, and high misclassification rates, providing a more convenient and effective solution. Furthermore, it offers a superior framework for multi-category and fine-grained image recognition. The anticipated applications of this method are expected to enhance the accuracy and efficiency of rice seed classification, thereby contributing to advancements in rice research and agricultural production. The structure of the article is arranged as shown in Fig 1.

## Materials and methods

### Dataset creation

In this experiment, a high-definition industrial electron microscope from the Leyue brand was utilized to collect the experimental dataset (Fig 2). To simulate the daily rice trading and laboratory conditions found in agricultural research institutes, LED lighting was used to provide supplementary illumination during the experiment, ensuring consistent brightness, focal length, and shooting depth for each seed. The seeds were systematically arranged on a black acrylic board to achieve the highest quality of experimental data, with output images saved in JPG format at a resolution of 1280×720 pixels. Throughout the data collection process, both high-quality and low-quality images were captured, including those featuring moldy seeds, noisy backgrounds, and dust contamination (Fig 3). Following data collection, researchers at the agricultural research institute carefully screened the dataset to ensure it was free from any human interference. The process of collecting data samples is shown in Fig 4.

Thirty-six rice seed varieties from different regions of China were thoughtfully selected for this study. Among these, 11 commercially promoted varieties, such as Heyouwuxiang, Taihexiangnuo, and Lijingyou 221, are featured. The remaining 25 varieties were sourced from a specific agricultural research institute in China, including D44, D45, and others (illustrated in Fig 5). For each variety, 390 surface feature images were captured, culminating in a comprehensive dataset of 14040 images. These images were organized into three sets: 300 for training, 40 for validation, and 50 for testing. It is noteworthy that none of the rice varieties used in this research have been previously studied in the context of deep learning for rice seed classification. To thoroughly evaluate the recognition accuracy of the developed network model, several genetically related rice seeds from the same region were selected, facilitating a more precise and realistic assessment of the model's performance.

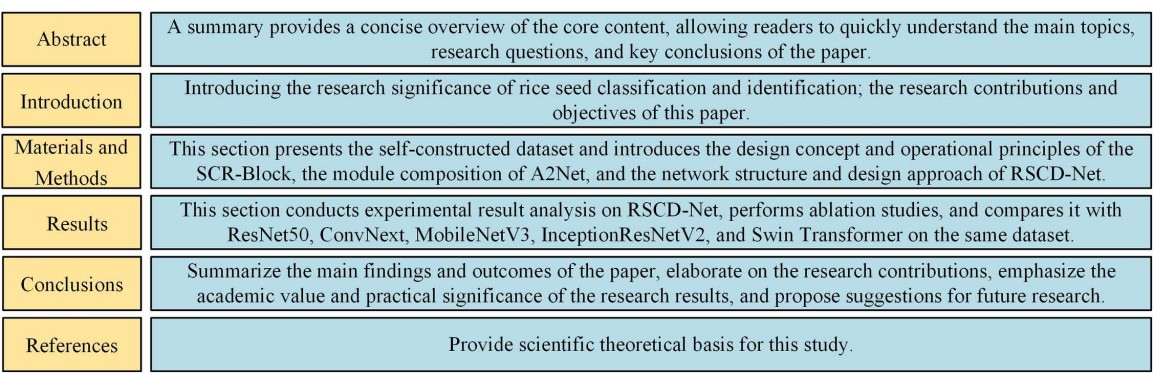

**Fig 1. The structure of the article is arranged.**

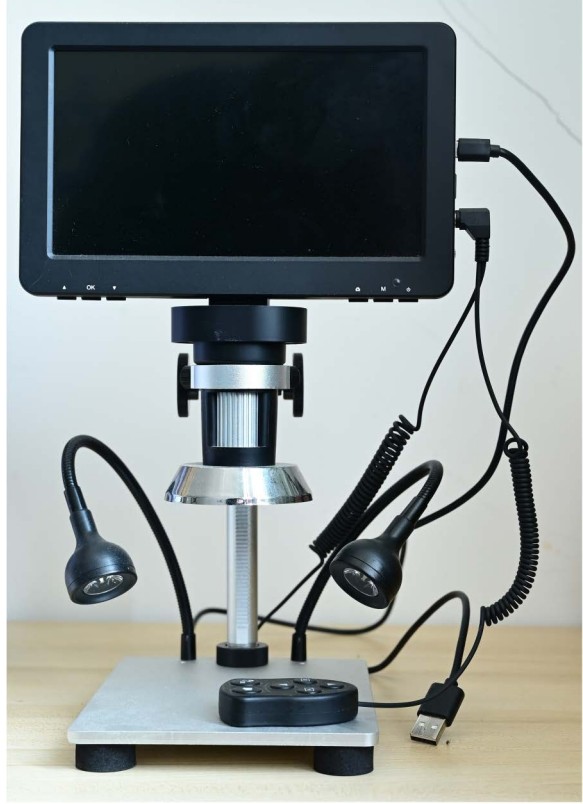

**Fig 2. Leyue HD industrial electron microscope.**

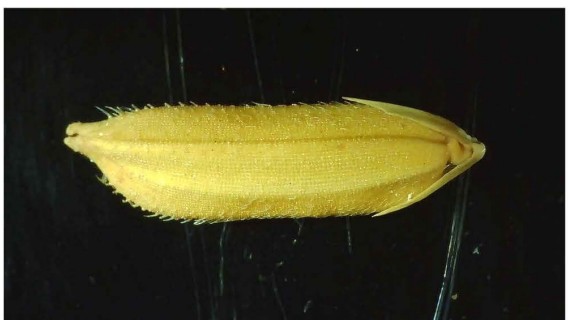 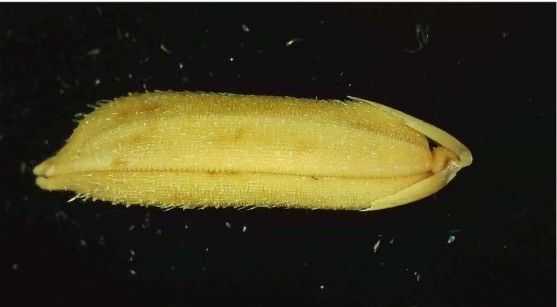

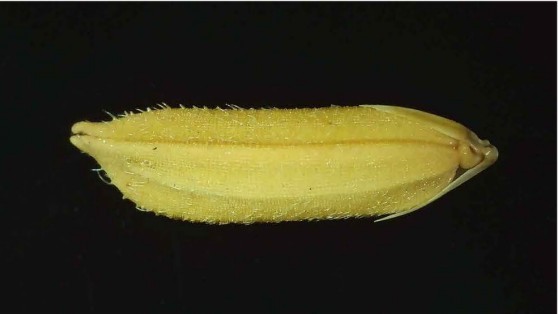

**Fig 3. Illustration of images with and without interference.**

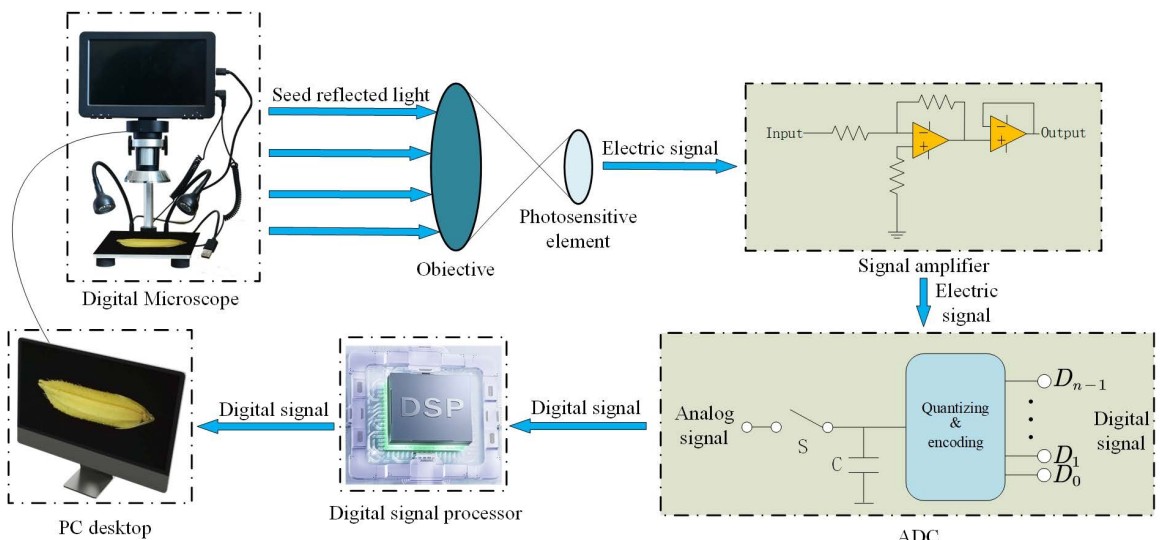

**Fig 4. Illustration of images with and without interference.**

## Experimental methods

Due to the specific variations among different rice seed types—including subtle differences in size, lemma hair length, and color within the same species—this type of image classification challenge is referred to as fine-grained recognition. Traditional deep learning networks often fall short of achieving the necessary recognition accuracy in this domain. To address this issue and propose a more effective model for multi-category fine-grained recognition, this study introduces an innovative rice seed identification network. This model is designed to overcome the challenges presented by small seed sizes, subtle inter-variety differences, and the intricacies of fine-grained recognition.

**Space and channel feature extraction residual block (SCR-Block).** Throughout the evolution of machine learning, various model compression strategies and advanced network design methods have emerged, including network pruning [20], weight quantization [21], low-rank factorization [22], and knowledge distillation [23]. However, these techniques are primarily regarded as post-processing optimizations for existing models, often limiting their performance by the upper bounds of the initial model. To more effectively extract features from the surface images of rice seeds, this study presents a lightweight Space and channel Feature Extraction Residual Network (SCR-Block), as depicted in Fig 6. The SCR-Block brings significant enhancements to space and channel strategies while incorporating a residual network to mitigate the vanishing gradient problem, thereby optimizing network performance and bolstering model stability and robustness. The core objective of the SCR-Block is to effectively reduce redundant information within feature data, leading to a substantial decrease in both model parameters and Floating Point Operations (FLOPs), while simultaneously enhancing the model's feature expression capabilities. By exploring space and channel redundancies among features, it successfully diminishes feature redundancy and improves the backbone network's feature extraction efficiency. This approach not only reduces computational complexity but also significantly enhances the network's accuracy and operational efficiency.

The SCR-Block comprises two primary components: the Spatial Reconstruction Unit (SRU) and the Channel Reconstruction Unit (CRU) [24]. The SRU implements an innovative "separate-reconstruct" approach to effectively minimize spatial redundancy in feature maps, optimizing the model's efficiency in processing spatial information. In parallel, the CRU utilizes a "segmentation-conversion-fusion" strategy to reduce redundancy in the channel dimension, enhancing the network's efficiency at that level.

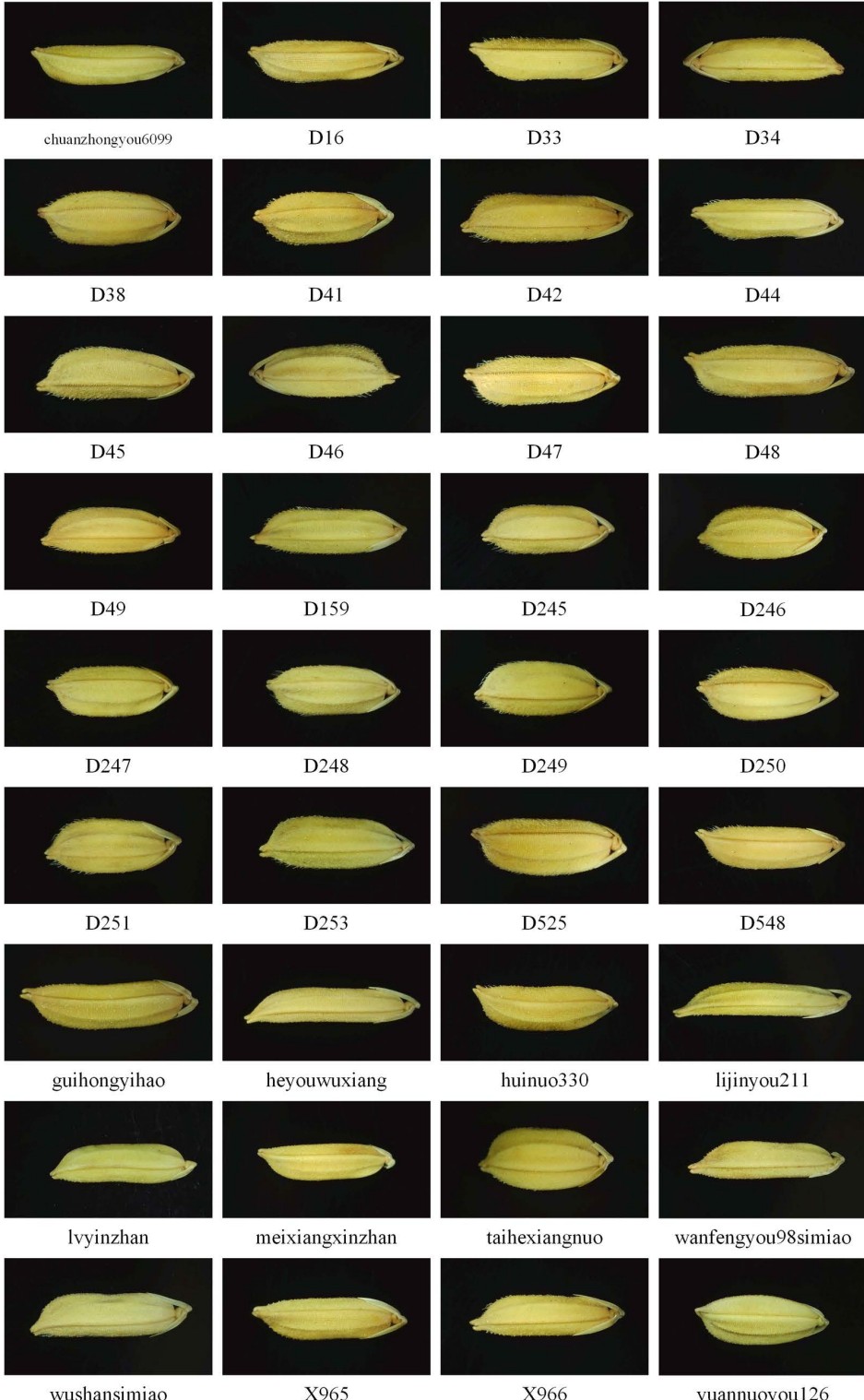

**Fig 5. Samples of 36 rice seeds.**

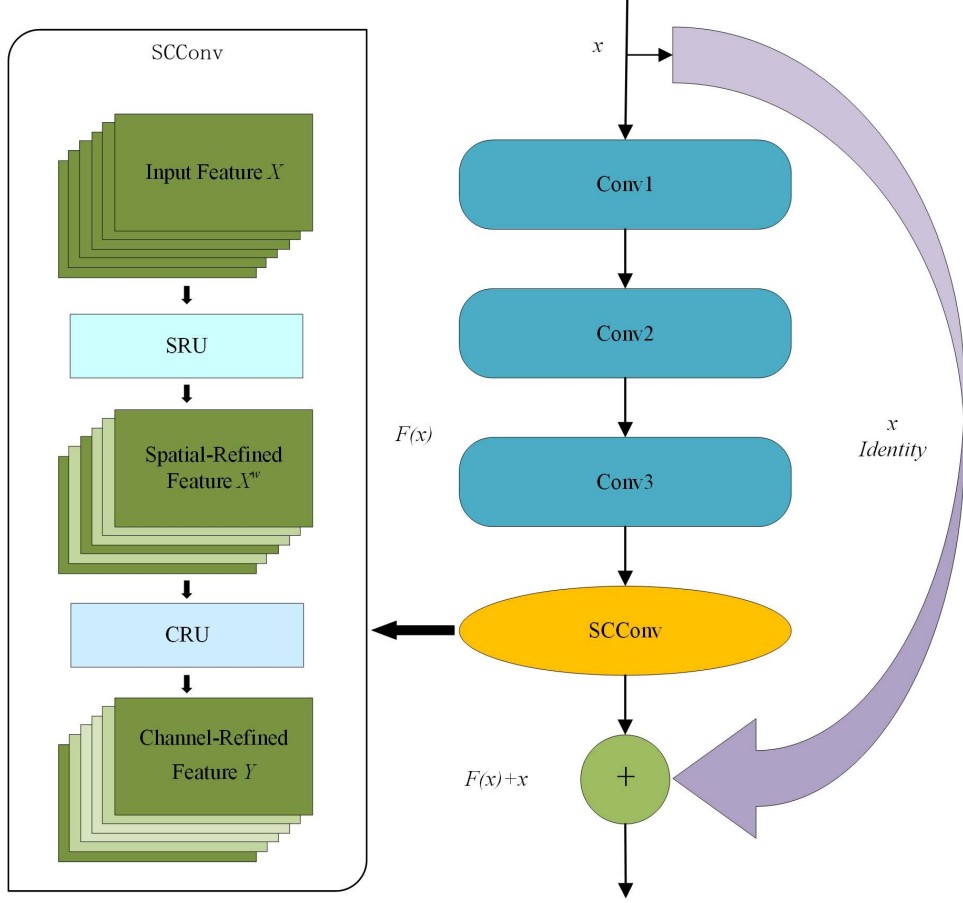

**Fig 6. Space and channel feature extraction residual block (SCR-Block).**

The Spatial Reconstruction Unit (SRU) is a mechanism that capitalizes on the spatial redundancy of features for enhanced processing. It operates through a two-step process: separation and reconstruction, as illustrated in Fig 7. During the separation phase, the aim is to distinguish between feature maps that convey rich information and those that exhibit lower spatial content correlation and minimal informative value. To evaluate the information content in various feature maps, the SRU employs the scaling factors derived from the Group Normalization (GN) layer. For an intermediate feature map $X \in R^{N \times C \times H \times W}$, N is the batch dimension, C is the channel dimension, and H and W are the spatial height and width. the SRU initiates the process by normalizing the input feature X. This normalization involves two steps: first, the mean value $\mu$ for each channel is subtracted, and then the result is divided by the standard deviation $\sigma$. This is shown in Equation (1):

$$X_{out} = GN(x) = \gamma \frac{X - \mu}{\sqrt{\sigma^2 + \varepsilon}} + \beta$$

(1)

In this context, $\mu$ and $\sigma$ denote the mean and standard deviation of the random variable $X$, respectively. Epsilon $\varepsilon$ is a small positive constant incorporated to enhance the algorithm's numerical stability. Furthermore, the parameters gamma $\gamma$ and $\beta$ are trainable within the layer, allowing for the definition of an affine transformation. Within the Group Normalization (GN) layer, the trainable parameter $\gamma \in R^C$ is responsible for quantifying the spatial pixel variance per batch and channel.

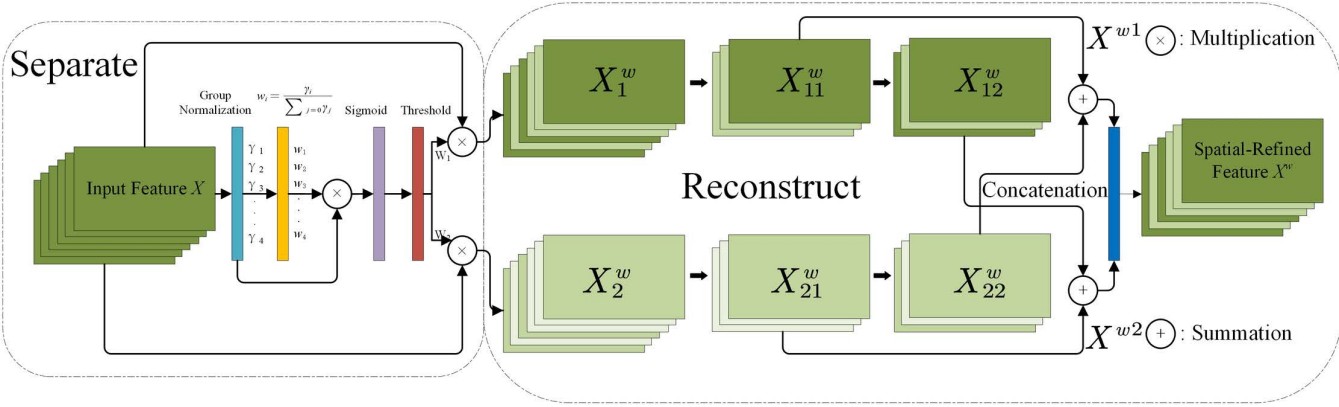

**Fig 7. Spatial reconstruction unit (SRU).**

The value of $\gamma$ increases with the richness of spatial information in the feature map and the variation among spatial pixels. Simultaneously, the normalization weight $W_\gamma \in R^C$ can be derived from Equation (2) and illustrates the relative significance of different feature maps.

$$W_\gamma = \{w_i\} = \frac{\gamma_i}{\Sigma_{j=i}^{C}\gamma_i} \qquad i,j = 1,2\ldots C \tag{2}$$

In this process, the weighted feature map, denoted as $w_\gamma$, is transformed into the range of (0, 1) using the sigmoid function and then gated with a threshold (*Gate*). We set the weights that exceed the threshold to 1 to obtain the information weights, represented as $W_1$, while weights below the threshold are set to 0, resulting in the non-information weights, denoted as $W_2$. In the experiments, the threshold is set at 0.5. This entire procedure for obtaining $W$ can be summarized in Equation (3). By employing this method, the model effectively filters out features that are more relevant to the current task, while suppressing those that are less important, ultimately enhancing the model's performance and generalization capability.

$$W = Gate(Sigmoid(W_\gamma(GN(X)))) \tag{3}$$

The input feature $X$ is weighted separately using two sets of weights, $W_1$ and $W_2$, resulting in two weighted features with distinct information contents: $X_1^w$, which possesses a higher information content, and $X_2^w$, which contains a lower information content. This process effectively partitions the input features into two categories. On one side, $X_1^w$ is rich in information and expressiveness, effectively capturing the critical aspects of spatial content. Conversely, $X_2^w$ contains less information and primarily reflects redundancy within the features. Through this separation mechanism, the SRU emphasizes valuable spatial information while minimizing unnecessary redundancy, thereby enhancing the efficiency of processing and utilizing feature maps within the neural network.

In the reconstruction process, features with higher information content, referred to as $X^{w_1}$, are combined with those possessing lower information content, denoted as $X^{w_2}$. This amalgamation produces a feature enriched with information, which not only optimizes space but also enhances feature usage. To further encourage information exchange between these features, cross-reconstruction operations are utilized. These operations effectively integrate the weighted, distinctive informational features $X^{w_1}$ and $X^{w_2}$, thereby enhancing the information flow between them. The resulting cross-reconstructed features $X^{w_1}$ and $X^{w_2}$ are then concatenated to form a more refined spatial feature map. This mapping

incorporates information from various features, enabling it to capture more detailed and comprehensive visual content, while ultimately providing high-quality feature representations for subsequent image processing or analysis tasks.

The subsequent processes are as follows, from Equation (4) to Equation (8):

$$X_1^w = W_1 \otimes X \tag{4}$$

$$X_2^w = W_2 \otimes X \tag{5}$$

$$X_{11}^w \oplus X_{22}^w = X^{w_1} \tag{6}$$

$$X_{21}^w \oplus X_{12}^w = X^{w_2} \tag{7}$$

$$X^{w_1} \cup X^{w_2} = X^w \tag{8}$$

In this context, $\otimes$ signifies element-wise multiplication, $\oplus$ denotes element-wise summation, and $\cup$ refers to concatenation. When the Spatial Recurrent Unit (SRU) is applied to the intermediate input feature $X$, it effectively differentiates between informative and less informative features. This process reconstructs the features to enhance their representational quality while reducing redundancy in the spatial dimension. However, the refined spatial feature map $X^w$ still presents some redundancy in the channel dimension. Consequently, the SRU not only enhances the spatial expressiveness of the features but also optimizes their utilization efficiency.

The Channel Reconstruction Unit (CRU) addresses the issue of redundancy among feature channels in Convolutional Neural Networks (CNNs), as shown in Fig 8. Employing a sophisticated "separate-transform-merge" strategy, the CRU optimizes feature channels with precision. This approach effectively reduces the redundant information between channels, resulting in more efficient feature representation. Consequently, the CRU alleviates both the parameter load and computational requirements of the network, while simultaneously enhancing its feature extraction capabilities. Overall, the CRU streamlines the model structure and strengthens its representational power by refining feature channels. The structure of the CRU is depicted below.

The segmentation process divides the refined spatial feature $X^w$ into two distinct parts: one containing $\alpha C$ channels and the other containing $(1-\alpha)C$ channels. A $1*1$ convolution kernel is then applied to compress the channel numbers of both groups, resulting in $X_{up}$ and $X_{low}$. For the transformation operation, $X_{up}$ serves as the input for "rich feature extraction." This involves performing Global Weighted Context (GWC) and Pixel-Wise Context (PWC) operations separately, after which the results are combined to produce the output $Y_1$. Concurrently, $X_{low}$ acts as supplementary input for

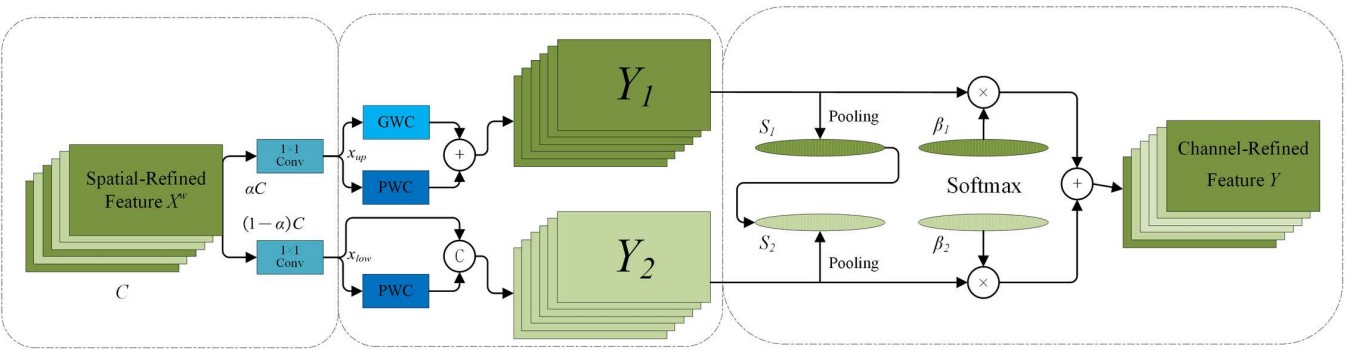

**Fig 8. Channel reconstruction unit (CRU).**

"rich feature extraction," undergoing PWC, and its output is integrated with the original input using a union operation to yield $Y_2$. In the fusion phase, a simplified Selective Kernel Network (SKNet) approach is employed to adaptively merge $Y_1$ and $Y_2$. The process begins with global average pooling to consolidate global space and channel statistical information, generating pooled values S1 and S2. SoftMax is then applied to both S1 and S2 to derive feature weight vectors $\beta_1$ and $\beta_2$. Finally, the output $Y$ is computed as follows: $Y = \beta_1 * Y_1 + \beta_2 * Y_2$ where $Y$ represents the refined channel feature.

**Double attention networks (A²Net).** Traditional methods for feature extraction typically involve calculating first-order statistics, such as average pooling and max pooling. In recent years, many advanced networks have begun to utilize bi-linear pooling to capture second-order statistics of features [25], thereby facilitating the generation of global representations within the network. Compared to traditional methods, bi-linear pooling is more effective in capturing feature capabilities and maintaining complex relationships. Given the subtlety of features among various types of rice seeds and the need for fine-grained distinctions, employing bi-linear pooling for feature extraction can significantly enhance the global receptive field. Bi-linear pooling provides all feature vector pairs ($a_i$; $b_i$) within two input feature maps A and B, as shown in formula (9).

$$G_{gather}(A, B) = AB^\top = \sum_{\forall i} a_i b_i^\top \tag{9}$$

A and B are the outputs of two different convolutional layers that transform input X, $A = \varphi(X; W_\varphi)$ and $B = softmax\left(\theta\left(X; W_\theta\right)\right)$.

After resolving the spatial feature aggregation issue, the next step involves distributing these spatial features to each position in the input. This distribution allows subsequent networks to capture global information with the use of smaller convolutional kernels. The A²Net network assigns adaptive visual primitives to the local feature vector $v_i$ at each position, leading to enhanced flexibility [26]. As a result, the features at each position can complement one another, making the training process more straightforward and improving the ability to capture complex relationships. As shown in Equation (10), $G_{gather}(x)$ employs soft attention to select a feature vector, enabling the network to effectively capture global information. Using the softmax function to normalize $v_i$ to a unit sum function, it was found that it has better convergence. Note that the set of weight vectors is also generated by convolutional layers and softmax normalizers, $V = softmax\left(\rho\left(X; W_\rho\right)\right)$, $W_\rho$ contains the parameters of this layer. Each position $i$ will generate its own attention vector based on the needs of its local feature $v_i$, and select a desired subset from the global features to help supplement the current position and form feature $z_i$.

$$z_i = \sum_{\forall j} v_{ij} g_j = G_{gather}(x) v_i, \, where \sum_{\forall j} v_{ij} = 1\# \tag{10}$$

The Double Attention Networks (A²Net) integrate a spatial feature aggregation module with a global information capture module. The computational model of A²Net, depicted in Fig 9, corresponds to the double attention operation outlined in Equation (11). The fundamental concept of A²Net involves employing an auxiliary network to guide the attention distribution of the main network. The auxiliary network first extracts features from the input image, subsequently generating an attention map that emphasizes the importance of various regions. The main network then utilizes this attention map to refine its attention distribution, enabling it to focus more intently on areas pertinent to the task at hand. A²Net significantly enhances the network's capability to capture both local key information and global contextual insights, effectively highlighting the crucial features within the image data. By incorporating this innovative network module, the model is better equipped to identify and process visual information with greater accuracy, resulting in a marked improvement in overall recognition performance.

$$\begin{aligned}
Z &= F_{distr}\left(G_{gather}(X); Y\right) \\
&= G_{gather}(X) softmax\left(\rho\left(X; W_\rho\right)\right) \\
&= \left[\varphi(X; W_\varphi) softmax\left(\theta\left(X; W_\theta\right)\right)^\top\right] softmax\left(\rho\left(X; W_\rho\right)\right)
\end{aligned} \tag{11}$$

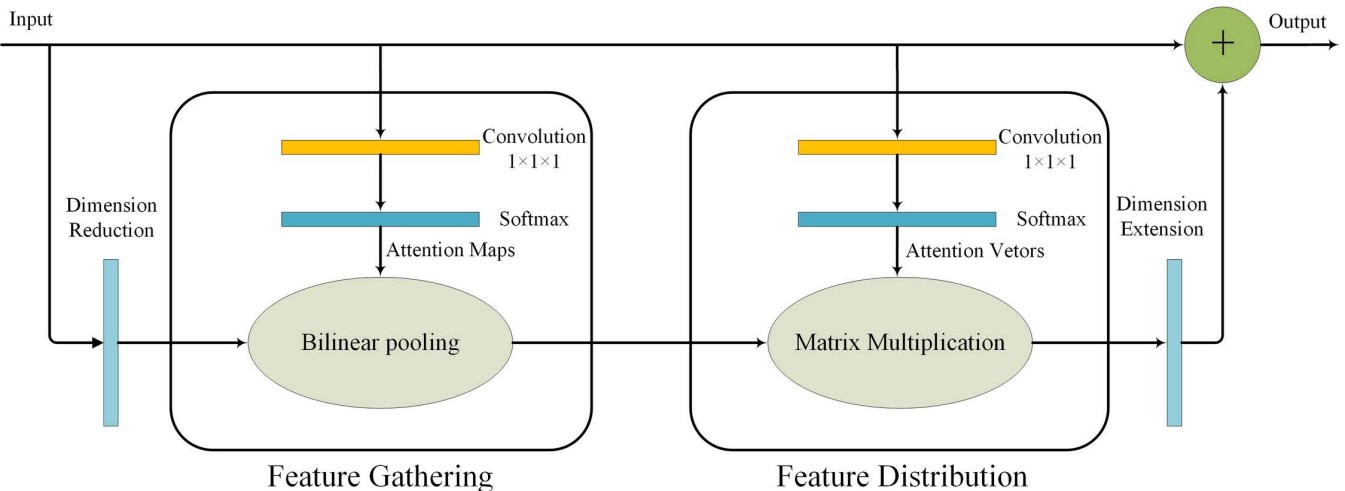

**Fig 9. Double attention networks (A²Net).**

## Deep space and channel residual network combined with double attention mechanism (RSCD-Net)

Due to the complexities involved in identifying rice species and the challenges posed by fine-grained recognition, this study presents a novel network model known as the Deep Space and Channel Residual Network Combined with Double Attention Mechanism (RSCD-Net), as depicted in Fig 10. The model parameters of d are shown in Fig 11. The model accepts an image of rice seeds as input and initiates feature extraction through a series of convolutional layers, batch normalization (BN) layers, ReLU activation functions, and max pooling operations. Following this initial phase, the model utilizes 16 layers of SCR-Block to extract crucial features, as well as space and channel information from the image. The resulting feature representations are then processed by the Double Attention Network (A²Net), which efficiently filters out the most significant features. The extraction process culminates in the network's average pooling layer. As demonstrated in the model's structure, the 16 layers of SCR-Block are organized into four primary convolutional blocks: the first block comprises 3 SCR-Blocks, the second includes 4 SCR-Blocks, the third consists of 6 SCR-Blocks, and the fourth again contains 3 SCR-Blocks.

## Results

### Experimental setup

We conducted experiments with 36 different varieties of rice seeds, utilizing a total of 14040 images, the specific division of the dataset is shown in Table 1. To maintain fairness and comparability in the experimental results, all network models were configured and executed under the same parameter settings, as detailed in Table 2. This methodology ensures an accurate assessment of each network's performance, thereby enhancing the scientific validity and reliability of the experimental findings. The original size of the images in the dataset is 1280x720. In order to reduce computational complexity while retaining the main information of the image, the network preprocessed the image and adjusted its size to 224x224.

### Analysis of experimental results

The performance of the RSCD-Net network was evaluated using a confusion matrix [27]. This network's design philosophy seamlessly integrates both space and channel convolutional neural network alongside residual learning. As a result, its experimental outcomes were compared with those of the classic baseline network, ResNet50,

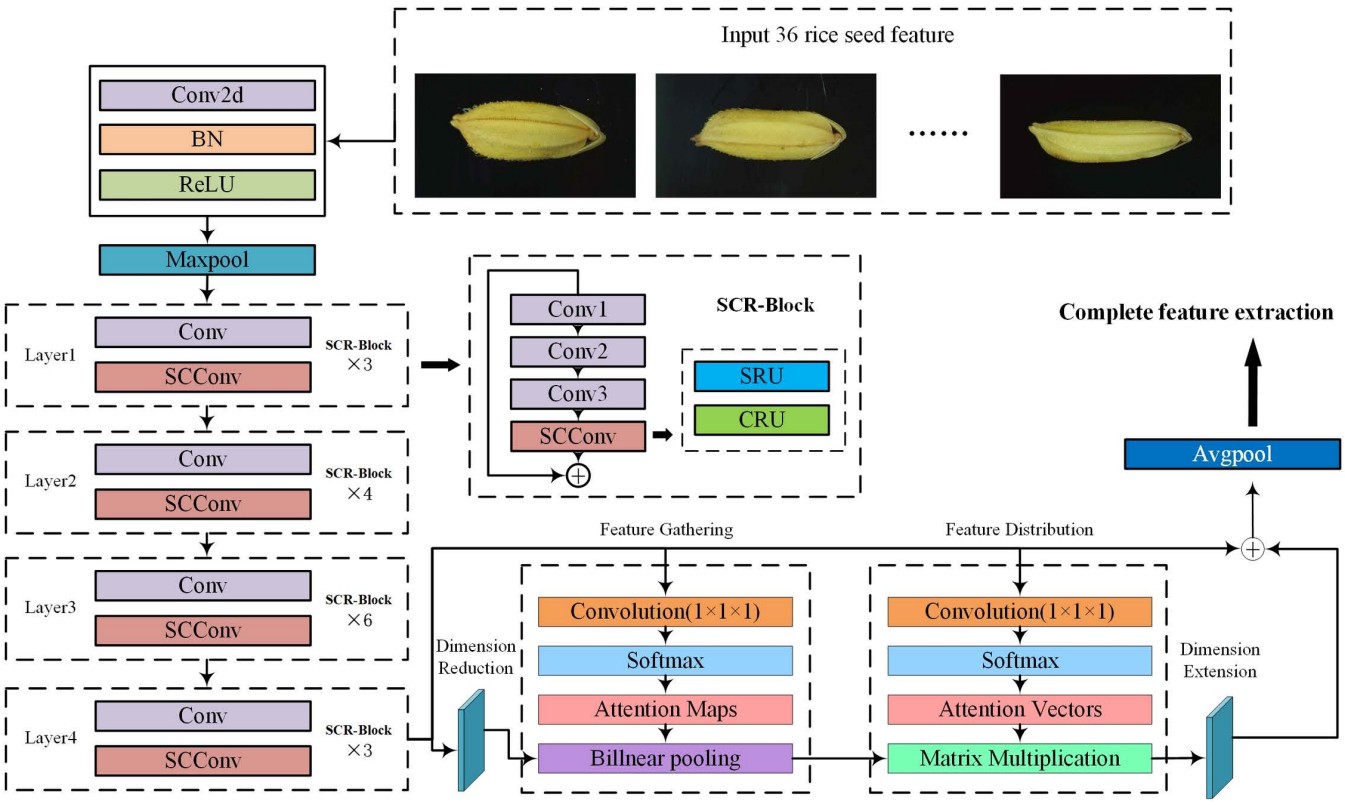

**Fig 10. Deep space and channel residual network combined with double attention mechanism.**

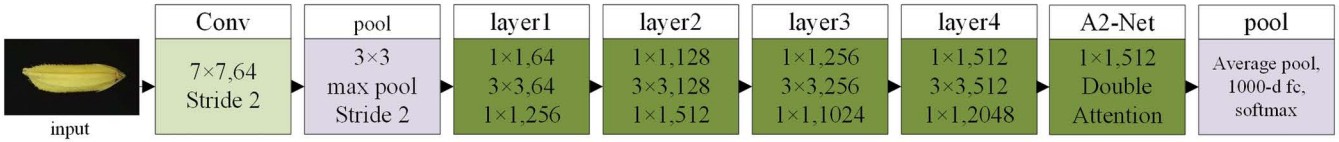

**Fig 11. Model parameters of RSCD-Net.**

**Table 1. Dataset partitioning.**

| Training set | 10800 |
|---|---|
| Validation set | 1440 |
| Test set | 1800 |
| Total | 12240 |

as shown in Figs 12 and 13. The visualization of the confusion matrix indicates that the RSCD-Net network's recognition performance is predominantly aligned along the diagonal, achieving an accuracy rate of 81.94%. This surpasses the 77.78% accuracy of the ResNet50 network, reflecting a notable improvement of 4.16%. For specific rice seed varieties, including D251, D33, D34, and D48, the accuracy rates increased by 20%, 22%, 10%, and 40%, respectively—all exceeding a minimal improvement of 0.1. Particularly noteworthy is variety D48, which

**Table 2. Fine tuning parameters.**

| Parameters | Values |
|---|---|
| Batch size | 32 |
| Number of epochs | 200 |
| optimizer | AdamW |
| Learning rate | 0.0005 |
| L2 regularization coefficient | 0.05 |
| Enter image size | 224*224 |

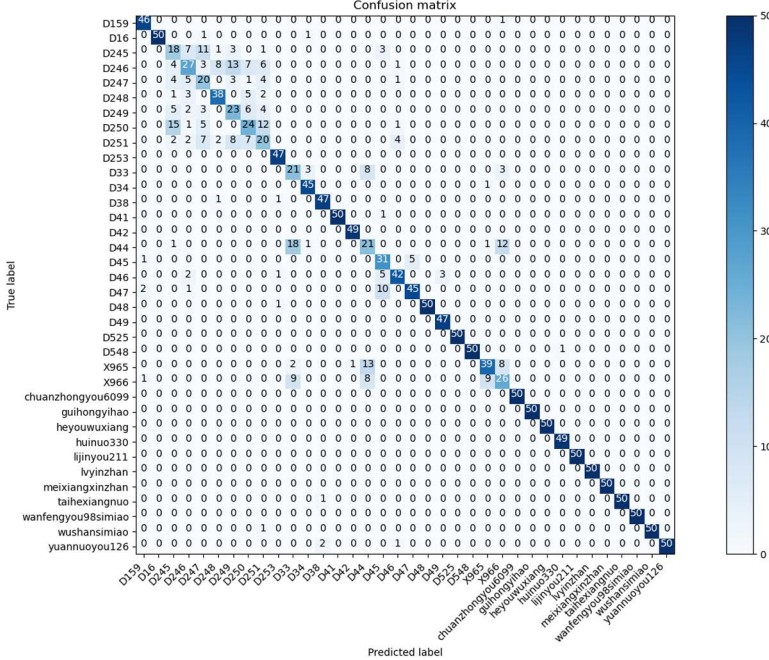

**Fig 12. Confusion Matrix of RSCD-Net.**

demonstrated the most significant enhancement, with the number of correctly identified samples increasing from 30 in the ResNet50 network to 50 in the RSCD-Net network, resulting in a remarkable 40% boost in accuracy. Furthermore, 15 varieties achieved a perfect recognition accuracy with the RSCD-Net network—four more than with the ResNet50 network. Despite the diversity among rice seed varieties and the subtle differences between samples, some varieties had less-than-optimal accuracy rates when assessed with the ResNet50 network. However, following feature extraction using the RSCD-Net, the number of correctly identified samples for these varieties significantly improved. For example, the number of correct identifications for variety D251 increased from 10 in the ResNet50 network to 20 in the RSCD-Net network, leading to an 20% enhancement in accuracy. In the case of variety D33, the number of correct identifications grew from 10 in the ResNet50 network to 21 in the RSCD-Net network, marking a 22% improvement in accuracy.

The results clearly demonstrate that the RSCD-Net network excels in extracting features from multi-variety, fine-grained samples, significantly outperforming the traditional ResNet50 network in both accuracy and robustness.

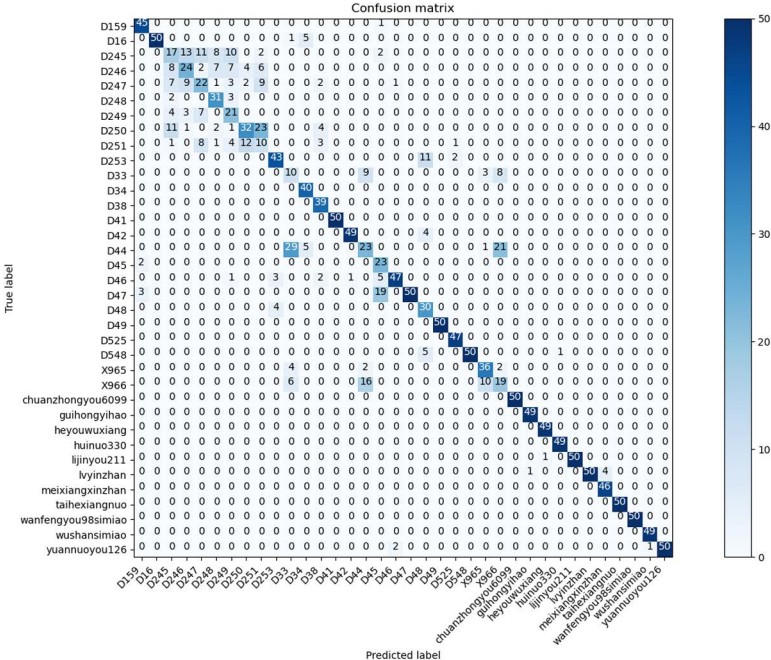

**Fig 13. Confusion matrix of ResNet50.**

## Ablation experiment

To present model performance more intuitively and scientifically, this paper employs Average Accuracy (A), Precision (P), Recall (R), and F1-score (F) as metrics for evaluating classification performance, with their respective formulas outlined in Equations (12) through (15). To maintain scientific rigor and fairness in assessment, all networks under comparison underwent five independent training sessions conducted under identical sample and environmental conditions. The weight files associated with the best-performing results were selected for sample identification. This approach ensures controlled variables and facilitates an equitable comparison of performance across all networks.

$$Average\ Accuracy = \frac{\sum TP}{N} \tag{12}$$

$$Precision = \frac{TP}{TP+FP} \tag{13}$$

$$Recall = \frac{TP}{TP+TN} \tag{14}$$

$$F1\_score = 2 \times \frac{Precision \times Recall}{Precision + Recall} \tag{15}$$

In this context, TP (True Positive) signifies instances where the actual class is positive, and the model accurately predicts it as such. TN (True Negative) represents cases where the actual class is negative, and the model correctly identifies it as negative. FP (False Positive) refers to situations in which the actual class is negative, yet the model incorrectly predicts it as positive. FN (False Negative) indicates instances where the actual class is positive, but the model erroneously predicts it as negative.

The RSCD-Net network was compared with the classic residual network ResNet50, an improved version of ResNet50 incorporating the SCR-Block module (denoted as ResNet50-SC), and another enhanced version integrating the A²Net

module (denoted as ResNet50-A$^2$) through ablation experiments to analyze the superiority of the improvements made in the RSCD-Net network.

The experimental results presented in Table 3 clearly indicate that RSCD-Net surpasses other comparative network models in accuracy, achieving the highest performance overall. The results of the ablation experiment data indicate that the effectiveness of the SCR-Block module in the base network for efficiently capturing features of the target data. This improvement is particularly beneficial for identifying different varieties of rice and yields superior results when handling large-scale sample datasets. On the other hand, ResNet50-A$^2$ achieves a 0.72% accuracy increase over ResNet50, suggesting that the A$^2$Net attention mechanism positively influences the control of local features, has a significant effect on improving the accuracy of fine-grained image recognition.

This article delves deeper into the results of the ablation experiment through visualized heat maps (as shown in Figs 14 and 15). The figures reveal that various networks effectively extract key features of rice seeds, including the edges of the lemma, overall size, and epidermal texture. Among these networks, RSCD-Net demonstrates the most concentrated attention distribution on rice seeds, followed by ResNet50-SC, ResNet50-A$^2$, and finally ResNet50. Notably, both ResNet50-SC and ResNet50 extract a significant number of features from non-subject regions. However, in comparison,

**Table 3. Comparison of experimental results of ablation models.**

| Comparison network | Accuracy | The difference with RSCD-Net |
| --- | --- | --- |
| ResNet50 | 77.78% | 4.16% |
| ResNet50-SC | 80.55% | 1.39% |
| ResNet50-A$^2$ | 79.11% | 2.83% |
| RSCD-Net | **81.94%** | |

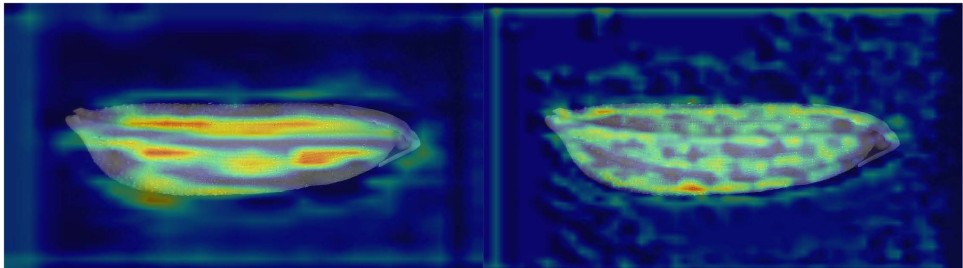

**Fig 14. Feature extraction heatmap of RSCD-Net (left) and ResNet50-SC (right).**

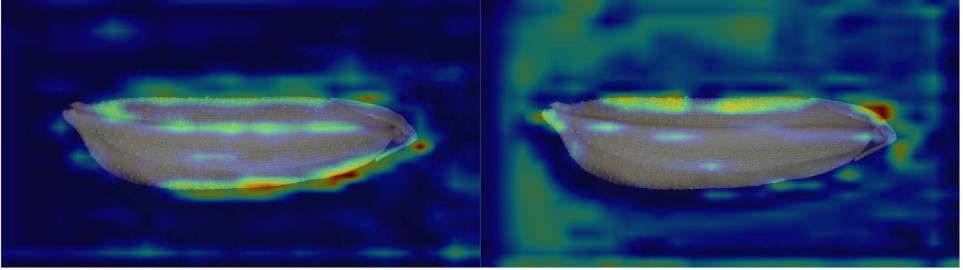

**Fig 15. Feature extraction heatmap of ResNet50-A$^2$ (left) and ResNet50 (right).**

ResNet50-SC not only captures features from non-subject areas but also achieves more comprehensive feature extraction from the main region of the rice seed. In contrast, ResNet50 exhibits limited feature extraction in the seed's primary area, indicating that the SCR-Block effectively enhances the network's capability to extract salient features from the image subject. Furthermore, ResNet50-A$^2$ excels in extracting the primary features of rice seeds while minimizing attention to non-subject regions. Compared to ResNet50, ResNet50-A$^2$ more precisely focuses its attention on the key subject of the image, demonstrating the effectiveness of A$^2$-Net in enhancing fine-grained image recognition accuracy. This also highlights its strong robustness and ability to improve feature extraction precision within deep learning models.

The experimental results clearly show that the combination of the SCR-Block module and A$^2$Net is very compatible and produces excellent integration effects, further validating the advantages and effectiveness of the RSCD-Net design.

### Comparison with other networks

To illustrate the performance of the RSCD-Net network, we conducted a comparison with several leading networks, including ResNet50 [28], ConvNext [29], MobileNetV3 [30], and InceptionResNetV2 [31], Swin Transformer [32], and MobileNetV3 [33], assessing them on metrics such as accuracy, precision, recall, and F1-score. The results can be found in Tables 4 and 5, where the top values for each metric are highlighted in bold and underlined. The confusion matrices of each model are shown in Figs 16 to 17.

Based on the data provided in the Tables 6 and 7, Figs 16 and 17, the new model RSCD-Net showcases exceptional performance in terms of accuracy (P), recall (R), F1-score (F), and average accuracy (A), effectively recognizing more data categories than the other three network models.

Comprehensively analyze the performance of each model on the four evaluation indicators, and the specific optimal values are shown in the Table 6:

Overall, RSCD-Net surpassed the performance of the other models. The recall rates (R) associated with various seeds reveal that RSCD-Net significantly outperformed the others, leading the field with 19 optimal metrics—four more than the second-place model, ConvNext. Additionally, all two networks demonstrated recognition accuracies of at least 18% for all rice seed varieties, underscoring their ability to capture subtle features that distinguish between different varieties. This validates the effectiveness of employing deep learning techniques for rice seed classification.

The evaluation metrics for the new model, RSCD-Net, indicate that its optimal values are predominantly found in categories that exhibit similar inter-species characteristics. This suggests the model's effectiveness in classifying rice seeds that closely resemble one another in appearance. In contrast, for rice seeds with more pronounced differences in their inter-species characteristics, the evaluation metrics among the various models show only minimal variation. According to the experimental data in Table 7, these findings that clearly underscore the significant advantages of RSCD-Net in addressing complex multi-category classification tasks that involve a variety of fine-grained rice seed identifications. This not only highlights the model's effectiveness in rice seed classification and recognition but also indirectly illustrates the limitations of lightweight networks in managing multi-variety, fine-grained image recognition tasks.

### Conclusions

This study presents a novel deep learning model, RSCD-Net, designed specifically for the classification and recognition of diverse rice seed varieties. By integrating the Space and Channel Residual Block (SCR-Block) and a Double Attention Mechanism (A$^2$Net), the proposed model effectively enhances feature extraction, minimizes redundant information, and optimizes computational efficiency. The backbone of RSCD-Net consists of 16 layers of SCR-Blocks, systematically arranged to maximize feature differentiation. Additionally, A$^2$Net expands the network's global receptive field, further improving its ability to distinguish between different rice seed varieties. To address overfitting and improve generalization, data augmentation techniques were applied during training, increasing the diversity of the dataset.

**Table 4. Comparison of experimental results of each model (1).**

| Name | RSCD-Net | | | | ResNet50 | | | | ConvNext | | | |
|---|---|---|---|---|---|---|---|---|---|---|---|---|
| | P | R | F | A | P | R | F | A | P | R | F | A |
| D159 | **0.979** | 0.92 | **0.949** | 0.767 | 0.978 | 0.9 | 0.937 | 0.75 | 0.958 | 0.92 | 0.939 | 0.767 |
| D16 | **0.962** | **1** | **0.981** | **0.833** | 0.893 | **1** | 0.943 | **0.833** | 0.96 | 0.96 | 0.96 | 0.8 |
| D245 | **0.409** | 0.36 | 0.383 | 0.3 | 0.27 | 0.34 | 0.301 | 0.283 | 0.333 | **0.46** | 0.386 | **0.383** |
| D246 | 0.391 | 0.54 | **0.454** | 0.45 | **0.414** | 0.48 | 0.445 | 0.4 | 0.325 | 0.52 | 0.4 | 0.433 |
| D247 | 0.526 | 0.4 | 0.454 | 0.333 | 0.393 | 0.44 | 0.415 | 0.367 | 0.554 | **0.62** | **0.585** | **0.517** |
| D248 | 0.776 | **0.76** | **0.768** | **0.633** | 0.861 | 0.62 | 0.721 | 0.517 | 0.683 | 0.56 | 0.615 | 0.467 |
| D249 | 0.535 | 0.46 | 0.495 | 0.383 | 0.6 | 0.42 | 0.494 | 0.35 | 0.6 | 0.42 | 0.494 | 0.35 |
| D250 | 0.414 | 0.48 | 0.445 | 0.4 | 0.432 | **0.64** | **0.516** | **0.533** | 0.438 | 0.42 | 0.429 | 0.35 |
| D251 | 0.385 | **0.4** | **0.392** | **0.333** | 0.25 | 0.2 | 0.222 | 0.167 | 0.407 | 0.22 | 0.286 | 0.183 |
| D253 | **1** | 0.94 | **0.969** | 0.783 | 0.768 | 0.86 | 0.811 | 0.717 | 0.862 | **1** | 0.926 | **0.833** |
| D33 | 0.6 | 0.42 | 0.494 | 0.35 | 0.333 | 0.2 | 0.25 | 0.167 | 0.69 | **0.8** | 0.741 | **0.667** |
| D34 | 0.978 | 0.9 | 0.937 | 0.75 | **1** | 0.8 | 0.889 | 0.667 | **1** | **1** | **1** | **0.833** |
| D38 | 0.959 | **0.94** | **0.949** | **0.783** | **1** | 0.78 | 0.876 | 0.65 | 0.848 | 0.78 | 0.813 | 0.65 |
| D41 | 0.98 | **1** | 0.99 | **0.833** | **1** | **1** | **1** | **0.833** | **1** | **1** | **1** | **0.833** |
| D42 | **1** | 0.98 | **0.99** | 0.817 | 0.925 | 0.98 | 0.952 | 0.817 | 0.926 | **1** | 0.962 | **0.833** |
| D44 | 0.389 | 0.42 | 0.404 | 0.35 | 0.291 | **0.46** | 0.356 | **0.383** | 0.792 | 0.38 | **0.514** | 0.317 |
| D45 | 0.838 | 0.62 | 0.713 | 0.517 | **0.92** | 0.46 | 0.613 | 0.383 | 0.9 | 0.54 | 0.675 | 0.45 |
| D46 | 0.792 | 0.84 | 0.815 | 0.7 | 0.797 | **0.94** | **0.863** | **0.783** | **0.85** | 0.68 | 0.756 | 0.567 |
| D47 | **0.776** | 0.9 | **0.833** | 0.75 | 0.694 | **1** | 0.819 | **0.833** | 0.662 | 0.94 | 0.777 | 0.783 |
| D48 | **0.98** | **1** | **0.99** | **0.833** | 0.882 | 0.6 | 0.714 | 0.5 | 0.977 | 0.84 | 0.903 | 0.7 |
| D49 | **1** | 0.94 | 0.969 | 0.783 | **1** | **1** | **1** | **0.833** | 0.98 | **1** | 0.99 | **0.833** |
| D525 | **1** | **1** | **1** | **0.833** | **1** | 0.94 | 0.969 | 0.783 | **1** | **1** | **1** | **0.833** |
| D548 | 0.98 | **1** | **0.99** | **0.833** | 0.893 | **1** | 0.943 | **0.833** | 0.98 | **1** | **0.99** | **0.833** |
| X965 | 0.619 | 0.78 | 0.69 | 0.65 | **0.818** | 0.72 | **0.766** | 0.6 | 0.46 | 0.8 | 0.584 | 0.667 |
| X966 | 0.491 | **0.52** | **0.505** | **0.433** | 0.373 | 0.38 | 0.376 | 0.317 | 0.545 | 0.36 | 0.434 | 0.3 |
| chuanzhongyou6099 | **1** | **1** | **1** | **0.833** | **1** | **1** | **1** | **0.833** | 0.98 | **1** | 0.99 | **0.833** |
| guihong1hao | **1** | **1** | **1** | **0.833** | **1** | 0.98 | 0.99 | 0.817 | **1** | **1** | **1** | **0.833** |
| heyouwuxiang | **1** | **1** | **1** | **0.833** | **1** | 0.98 | 0.99 | 0.817 | **1** | **1** | **1** | **0.833** |
| huinuo330 | **1** | 0.98 | **0.99** | 0.817 | **1** | 0.98 | **0.99** | 0.817 | **1** | 0.98 | **0.99** | 0.817 |
| lijinyou211 | **1** | **1** | **1** | **0.833** | 0.98 | **1** | 0.99 | **0.833** | **1** | **1** | **1** | **0.833** |
| lvyinzhan | **1** | **1** | **1** | **0.833** | 0.909 | **1** | 0.952 | **0.833** | **1** | 0.98 | 0.99 | 0.817 |
| meixiangxinzhan | **1** | **1** | **1** | **0.833** | **1** | 0.92 | 0.958 | 0.767 | **1** | **1** | **1** | **0.833** |
| taihexiangnuo | 0.98 | **1** | 0.99 | **0.833** | **1** | **1** | **1** | **0.833** | 0.961 | 0.98 | 0.97 | 0.817 |
| wanfengyou98simiao | **1** | **1** | **1** | **0.833** | **1** | **1** | **1** | **0.833** | 0.926 | **1** | 0.962 | **0.833** |
| wushansimiao | 0.98 | **1** | **0.99** | **0.833** | **1** | 0.98 | **0.99** | 0.817 | **1** | 0.92 | 0.958 | 0.767 |
| yuannuoyou126 | 0.943 | **1** | 0.971 | **0.833** | 0.943 | **1** | 0.971 | **0.833** | **1** | **1** | **1** | **0.833** |

Experimental validation was conducted on a self collected dataset that has not been previously used in deep learning-based rice seed classification. Despite its relatively small sample size compared to datasets in other related studies, RSCD-Net achieved an average accuracy of 81.94%, outperforming ResNet50 by 4.16% (77.78%). Furthermore, RSCD-Net demonstrated a significant improvement in recognition rates across a broader spectrum of rice seed variet- ies, clearly surpassing the performance of baseline models. To thoroughly evaluate the model's effectiveness, extensive ablation experiments and comparisons with leading architectures, including InceptionResNetV2 and ConvNeXt, were

Table 5. Comparison of experimental results of each model (2).

| Name | InceptionResnetV2 | | | | Swin Transformer | | | | MobileNetV3 | | | |
|---|---|---|---|---|---|---|---|---|---|---|---|---|
| | P | R | F | A | P | R | F | A | P | R | F | A |
| D159 | 0.909 | 0.8 | 0.851 | 0.667 | 0.839 | **0.94** | 0.887 | **0.783** | 0.907 | 0.78 | 0.839 | 0.65 |
| D16 | 0.653 | 0.94 | 0.771 | 0.783 | 0.657 | 0.88 | 0.752 | 0.733 | 0.571 | 0.64 | 0.604 | 0.533 |
| D245 | 0.39 | **0.46** | **0.422** | **0.383** | 0.389 | 0.28 | 0.326 | 0.233 | 0.18 | 0.18 | 0.18 | 0.15 |
| D246 | 0.294 | **0.6** | 0.395 | **0.5** | 0.144 | 0.28 | 0.19 | 0.233 | 0.227 | 0.34 | 0.272 | 0.283 |
| D247 | **0.556** | 0.6 | 0.577 | 0.5 | 0.315 | 0.56 | 0.403 | 0.467 | 0.25 | 0.2 | 0.222 | 0.167 |
| D248 | **0.857** | 0.6 | 0.706 | 0.5 | 0.7 | 0.28 | 0.4 | 0.233 | 0.485 | 0.32 | 0.386 | 0.267 |
| D249 | **0.686** | **0.48** | **0.565** | **0.4** | 0.378 | 0.28 | 0.322 | 0.233 | 0.364 | 0.24 | 0.289 | 0.2 |
| D250 | **0.556** | 0.3 | 0.39 | 0.25 | 0.442 | 0.38 | 0.409 | 0.317 | 0.266 | 0.42 | 0.326 | 0.35 |
| D251 | 0.52 | 0.26 | 0.347 | 0.217 | **0.826** | 0.38 | 0.521 | 0.317 | 0.189 | 0.2 | 0.194 | 0.167 |
| D253 | 0.926 | **1** | 0.962 | **0.833** | 0.917 | 0.88 | 0.898 | 0.733 | 0.733 | 0.66 | 0.695 | 0.55 |
| D33 | **0.83** | 0.78 | **0.804** | 0.65 | 0.597 | 0.74 | 0.661 | 0.617 | 0.447 | 0.76 | 0.563 | 0.633 |
| D34 | **1** | 0.9 | 0.947 | 0.75 | 0.88 | 0.88 | 0.88 | 0.733 | 0.65 | 0.52 | 0.578 | 0.433 |
| D38 | 0.844 | 0.76 | 0.8 | 0.633 | 0.71 | 0.88 | 0.786 | 0.733 | 0.625 | 0.6 | 0.612 | 0.5 |
| D41 | **1** | **1** | **1** | **0.833** | 1 | 0.98 | 0.99 | 0.817 | 0.619 | 0.78 | 0.69 | 0.65 |
| D42 | 0.814 | 0.96 | 0.881 | 0.8 | 0.804 | 0.82 | 0.812 | 0.683 | 0.574 | 0.62 | 0.596 | 0.517 |
| D44 | **0.846** | 0.22 | 0.349 | 0.183 | 0.607 | 0.34 | 0.436 | 0.283 | 0.176 | 0.06 | 0.089 | 0.05 |
| D45 | 0.679 | **0.76** | **0.717** | **0.633** | 0.6 | 0.36 | 0.45 | 0.3 | 0.545 | 0.48 | 0.51 | 0.4 |
| D46 | 0.772 | 0.88 | 0.822 | 0.733 | 0.522 | 0.48 | 0.5 | 0.4 | 0.333 | 0.46 | 0.386 | 0.383 |
| D47 | 0.755 | 0.8 | 0.777 | 0.667 | 0.642 | 0.68 | 0.66 | 0.567 | 0.491 | 0.54 | 0.514 | 0.45 |
| D48 | 0.768 | 0.86 | 0.811 | 0.717 | 0.667 | **1** | 0.8 | **0.833** | 0.7 | 0.42 | 0.525 | 0.35 |
| D49 | 0.935 | 0.86 | 0.896 | 0.717 | 0.722 | 0.78 | 0.75 | 0.65 | 0.673 | 0.66 | 0.666 | 0.55 |
| D525 | **1** | 0.92 | 0.958 | 0.767 | 1 | 1 | 1 | **0.833** | 0.923 | 0.72 | 0.809 | 0.6 |
| D548 | **1** | 0.94 | 0.969 | 0.783 | 0.554 | 0.82 | 0.661 | 0.683 | 0.612 | 0.6 | 0.606 | 0.5 |
| X965 | 0.396 | **0.84** | 0.538 | **0.7** | 0.486 | 0.7 | 0.574 | 0.583 | 0.455 | 0.5 | 0.476 | 0.417 |
| X966 | **0.688** | 0.22 | 0.333 | 0.183 | 0.364 | 0.16 | 0.222 | 0.133 | 0.444 | 0.24 | 0.312 | 0.2 |
| chuanzhongyou6099 | 0.926 | **1** | 0.962 | **0.833** | 1 | 0.76 | 0.864 | 0.633 | 0.772 | 0.88 | 0.822 | 0.733 |
| guihong1hao | **1** | **1** | **1** | **0.833** | 1 | 0.96 | 0.98 | 0.8 | 0.913 | 0.84 | 0.875 | 0.7 |
| heyouwuxiang | 0.92 | 0.92 | 0.92 | 0.767 | 0.86 | 0.98 | 0.916 | 0.817 | 0.792 | 0.76 | 0.776 | 0.633 |
| huinuo330 | 0.98 | **1** | **0.99** | **0.833** | 0.812 | 0.52 | 0.634 | 0.433 | 0.667 | 0.6 | 0.632 | 0.5 |
| lijinyou211 | 0.926 | **1** | 0.962 | **0.833** | 0.842 | 0.96 | 0.897 | 0.8 | 0.646 | 0.84 | 0.73 | 0.7 |
| lvyinzhan | **1** | 0.84 | 0.913 | 0.7 | 0.833 | 0.8 | 0.816 | 0.667 | 0.745 | 0.7 | 0.722 | 0.583 |
| meixiangxinzhan | **1** | **1** | **1** | **0.833** | 1 | 0.66 | 0.795 | 0.55 | 0.911 | 0.82 | 0.863 | 0.683 |
| taihexiangnuo | **1** | **1** | **1** | **0.833** | 0.877 | **1** | 0.934 | **0.833** | 0.831 | 0.98 | 0.899 | 0.817 |
| wanfengyou98simiao | 0.926 | **1** | 0.962 | **0.833** | 0.771 | 0.74 | 0.755 | 0..617 | 0.612 | 0.6 | 0.606 | 0.5 |
| wushansimiao | **1** | 0.92 | 0.958 | 0.767 | 0.805 | 0.66 | 0.725 | 0.55 | 0.833 | 0.8 | 0.816 | 0.667 |
| yuannuoyou126 | **1** | **1** | **1** | **0.833** | 1 | 0.94 | 0.969 | 0.783 | 0.7 | 0.84 | 0.764 | 0.7 |

Note: The optimal evaluation indicators are marked in bold and underlined.

conducted. The results confirm that RSCD-Net achieves superior overall performance, consistently exceeding the accuracy of both ablation and comparison models.

A comprehensive review of existing literature on rice seed classification reveals that most studies focus on no more than 10 seed varieties, significantly limiting their practical applications. This study makes a notable contribution by developing a model capable of accurately classifying 36 different rice seed types, including both widely cultivated varieties and

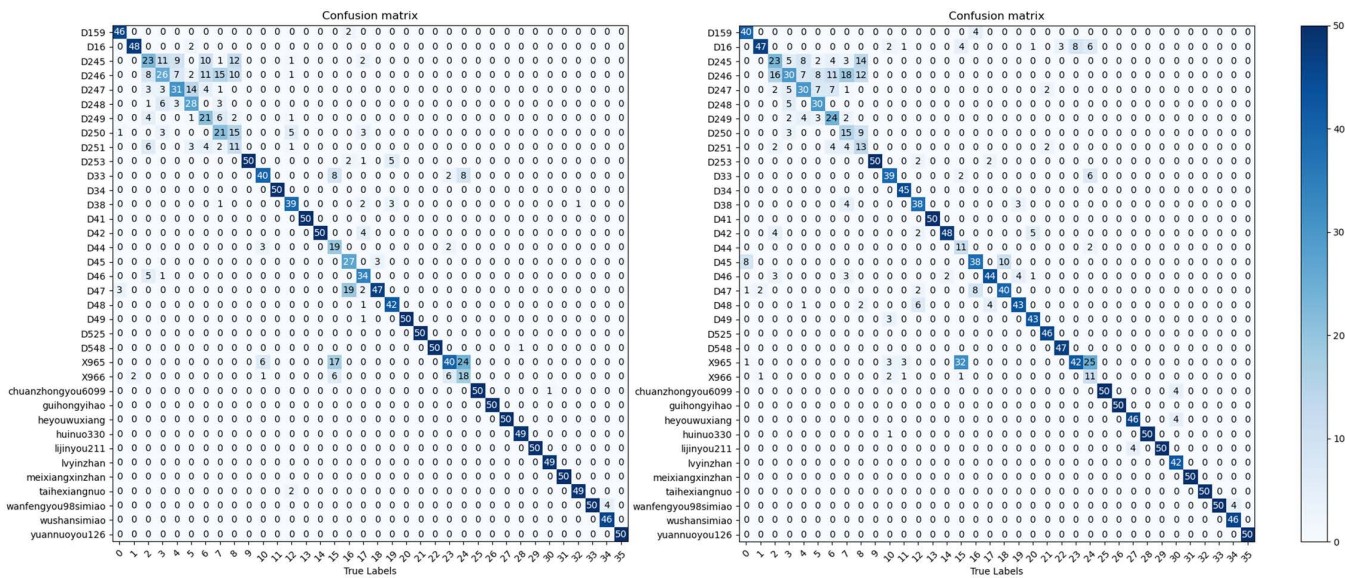

**Fig 16. Confusion Matrix of ConvNext(left) and InceptionResnetV2(right).**

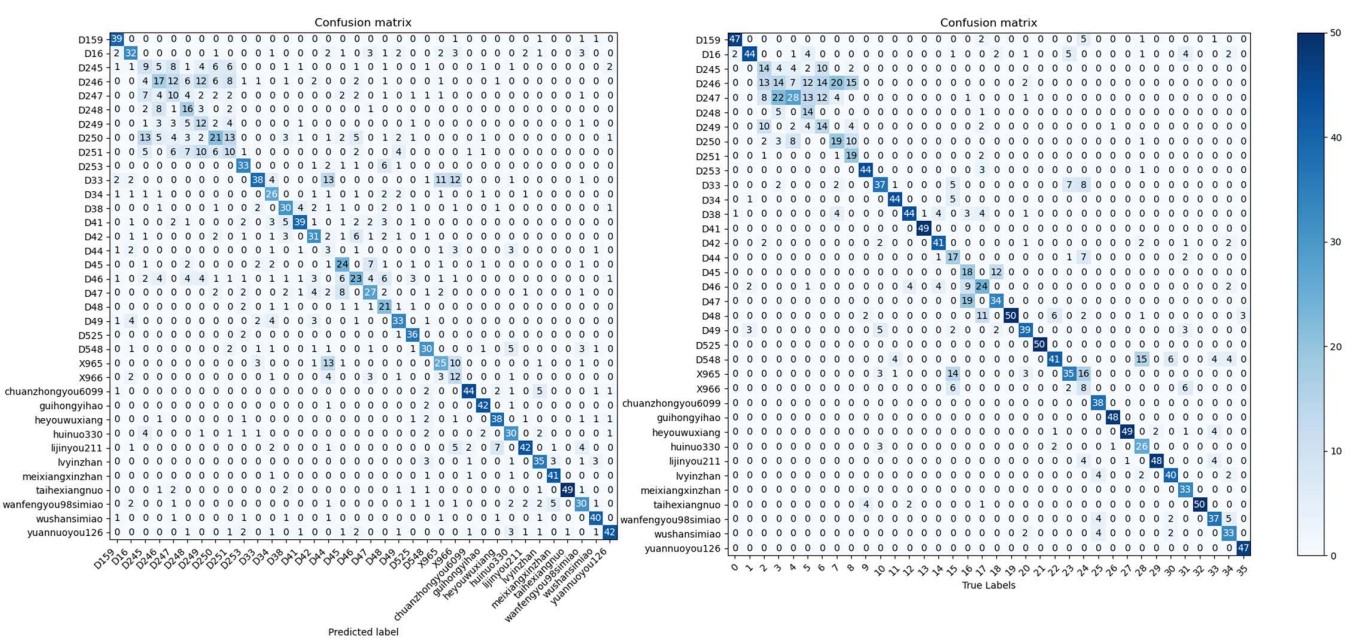

**Fig 17. Confusion Matrix of MobileNetV3(left) and Swin Transformer(right).**

specialized strains collected from an agricultural research institute. The exceptional classification capability of RSCD-Net offers practical benefits for farmers, rice traders, and researchers, providing a highly accurate, efficient, and cost-effective method for rice seed identification. Furthermore, the findings of this study lay the foundation for the development of a more comprehensive rice seed classification database, supporting future advancements in precision agriculture and smart farming technologies.

**Table 6. The optimal values achieved by each model under evaluation indicators P, R, F, and A.**

| Name | P | R | F | A |
|---|---|---|---|---|
| RSCD-Net | 17 | 19 | 22 | 19 |
| ResNet50 | 16 | 14 | 10 | 14 |
| ConvNext | 12 | 17 | 12 | 17 |
| InceptionResnetV2 | 17 | 15 | 10 | 15 |
| MobileNetV3 | 0 | 0 | 0 | 0 |
| Swin Transformer | 7 | 4 | 1 | 4 |

**Table 7. Comparison of recognition accuracy among different models.**

| Net | accuracy | The difference with RSCD-Net |
|---|---|---|
| RSCD-Net | **81.94%** | |
| ResNet50 | 77.78% | 4.16% |
| ConvNext | 80.77% | 1.17% |
| InceptionResnetV2 | 78.94% | 3% |
| MobileNetV3 | 57.22% | 24.72% |
| Swin Transformer | 68.72% | 13.22% |

In summary, RSCD-Net represents a powerful and scalable solution for fine-grained rice seed classification, demonstrating superior performance in both accuracy and efficiency. Its integration of advanced feature extraction mechanisms and attention-based learning strategies positions it as a valuable tool for both academic research and real-world applications in the agricultural sector. Future work will explore further enhancements to scalability, dataset expansion, and real-time deployment, ensuring broader adoption of RSCD-Net in agricultural automation and smart seed classification systems.

## Supporting information

**S1 File. Coded file.**
(ZIP)

## Author contributions

**Conceptualization:** Zhongyi Yang.

**Data curation:** Tingyuan Zhang, Dekai Li.

**Funding acquisition:** Meng Wang.

**Methodology:** Tingyuan Zhang, Dekai Li.

**Writing – original draft:** Tingyuan Zhang.

**Writing – review & editing:** Changsheng Zhang, Zhongyi Yang, Fujie Zhang, Dekai Li, Sen Yang.

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
