## [Decision Letter · Decision Letter 0]

21 Jan 2025

PONE-D-24-54192Multi-class Rice Seed Recognition Based on Deep Space and Channel Residual Network Combined with Double Attention MechanismPLOS ONE

Dear Dr. Changsheng,

Thank you for submitting your manuscript to PLOS ONE. After careful consideration, we feel that it has merit but does not fully meet PLOS ONE’s publication criteria as it currently stands. Therefore, we invite you to submit a revised version of the manuscript that addresses the points raised during the review process. The comments from revieweres are attached at the bottom of this email.

We look forward to receiving your revised manuscript.

Kind regards,

Xiaoyong Sun

Academic Editor

PLOS ONE

“This thesis supported by National Natural Science Foundation of China ( 62062048) and China Yunnan Province Science and Technology Plan Project(202201AT070113) 。”

Reviewers' comments:

Reviewer's Responses to Questions

**Comments to the Author**

1. Is the manuscript technically sound, and do the data support the conclusions?

Reviewer #1: Yes

Reviewer #2: Yes

2. Has the statistical analysis been performed appropriately and rigorously? 

Reviewer #1: Yes

Reviewer #2: N/A

3. Have the authors made all data underlying the findings in their manuscript fully available?

Reviewer #1: Yes

Reviewer #2: No

4. Is the manuscript presented in an intelligible fashion and written in standard English?

Reviewer #1: Yes

Reviewer #2: Yes

5. Review Comments to the Author

Reviewer #1: This paper “Multi-class Rice Seed Recognition Based on Deep Space and Channel Residual Network Combined with Double Attention Mechanism”, aims to to improve the recognition accuracy of 36 different types of rice seeds through the application of deep learning techniques, proposing a Deep Space and Channel Residual Network that combines double attention mechanism. To achieve this, the study introduces a Space and channel Feature Extraction Residual Block. This block is designed to enhance inter-class differences while minimizing redundant feature information, thereby effectively reducing the model’s computational complexity. The net consists of 16 layers of SCR-Blocks, incorporating batch normalization (BN) layers, max pooling layers, and average pooling layers. The SCRBlocks are organized into four main convolutional layers with 3, 4, 6, and 3 units, respectively, optimizing feature extraction. Additionally, the introduction of a double attention mechanism (Double Attention Networks, A2Net) improves the model’s global receptive field, enhancing its ability to distinguish between various seed varieties. Experiments conducted with a self-collected dataset revealed that the RSCD-Net model achieved an average accuracy of 82.61%, reflecting an improvement of 4.22% over the baseline model.

The topic is justified. The paper could be further improved if the following remarks are taken into consideration:

1. ABSTRACT: this needs to be re-written, summarize the methodology information and conclusion statements.

2. A few of the grammatical mistakes including punctuation etc. are found.

3. A good article may have keywords.

4. The introduction section may have contribution and objectives of the study key folded into it. The last paragraph of the introduction section may describe the layout of the of the rest of the draft.

5. JPG format at a resolution of 1280×720 for whole of the dataset processing seems a complex task, but authors mentioned that 224x224 image size is input?

6. Experimental setup: capital the first letter of we->We.

7. Learning rate, i.e., 0.0005, seems high.

8. Results are convincing.

9. Table 4, comparison is based on what? Did these models, i.e., RSCD-Net, ResNet50, ConvNext, InceptionResnetV2, and MobileNetV3 were trained on the same dataset?

10. The discussion needs to redraw under quantitative analysis.

11. The conclusion section should be comprehensive.

12. The motivation is not clear. Please specify the importance of the overall research activity.

Reviewer #2: The manuscript presents a novel deep learning model, RSCD-Net, for the classification of 36 rice seed varieties, addressing challenges in fine-grained image recognition. The study demonstrates innovation through the introduction of the Space and Channel Residual Block (SCR-Block) and Double Attention Mechanism (A2Net). These features improve the classification accuracy and efficiency, surpassing baseline models like ResNet50 and others.

Strengths:

1. Novelty: The RSCD-Net model incorporates innovative mechanisms (SCR-Block and A2Net) that address inter-class variability and redundancy in feature extraction.

2. Dataset: A self-collected dataset of 36 rice varieties is commendable, as such datasets are rare and critical for agricultural research.

3. Performance: The model achieves superior accuracy (82.61%) compared to established networks, with detailed experimental validations.

4. Scientific Rigor: Ablation studies and comparisons with multiple benchmarks provide a robust evaluation of the proposed model's effectiveness.

5. Practical Relevance: The study has significant implications for agricultural research and industrial applications, especially in seed sorting and quality assessment.

Weaknesses:

1. Dataset Limitations: While the dataset is unique, the sample size per variety (240 for training) is relatively small for deep learning, potentially limiting the model's generalizability.

2. Comparative Analysis: Although the model outperforms ResNet50 and similar architectures, a comparison with newer state-of-the-art methods like Vision Transformers (ViT) would strengthen the claim of superiority.

3. Reproducibility: The manuscript does not provide complete details for dataset access or pretrained model weights, which are essential for reproducibility.

4. Computational Efficiency: While RSCD-Net reduces redundancy, its computational cost compared to lightweight models like MobileNetV3 remains high, which may limit deployment in resource-constrained environments.

5. Clarity in Writing: The manuscript has some complex descriptions, especially in the Materials and Methods section, which could be streamlined for better understanding.

Recommendations:

1. Expand Dataset: Consider increasing the dataset size by using different augmentation techniques or incorporating external datasets for better generalization.

2. Broader Comparisons: Include experiments with state-of-the-art models like ViT or Swin Transformers for a more comprehensive evaluation.

3. Detailed Ablation: Provide additional insights into the contributions of SCR-Block and A2Net individually through detailed visualizations.

4. Provide Resources: Share dataset links, code repositories, or pretrained models to enhance reproducibility.

5. Improve Clarity: Simplify complex sections and add more diagrams or flowcharts to explain model architecture and methodology.

6. Improve the language: There are lot of spelling mistakes or typos. Correct them.

6. PLOS authors have the option to publish the peer review history of their article (what does this mean? ). If published, this will include your full peer review and any attached files.

**Do you want your identity to be public for this peer review?** For information about this choice, including consent withdrawal, please see our Privacy Policy .

Reviewer #1: **Yes: ** Ghulam Gilanie

Reviewer #2: No

---

## [Author Response · Author response to Decision Letter 1]

10 Mar 2025

Modification instructions

Dear PLoS ONE editorial team:

I am delighted to receive your letter and sincerely appreciate the recognition from you and the reviewers regarding our work in the article "Multi-Class Rice Seed Recognition Based on Deep Space and Channel Residual Network Combined with Double Attention Mechanism." We are also deeply grateful for the insightful suggestions provided to enhance our manuscript. Your feedback has been immensely valuable in refining and improving our paper, offering significant guidance for our research.

We fully acknowledge and agree with the recommendations made by you and the reviewers, and we have carefully revised the manuscript accordingly. To ensure clarity, we have marked the deleted sections in red and highlighted the modified or newly added content in blue within the text. A detailed explanation of the specific modifications is provided as follows:

一�Reply to PLoS ONE Journal Editor

1.Thank you for stating the following financial disclosure:

“This thesis supported by National Natural Science Foundation of China (62062048) and China Yunnan Province Science and Technology Plan Project(202201AT070113) .”Please state what role the funders took in the study. If the funders had no role, please state: "The funders had no role in study design, data collection and analysis, decision to publish, or preparation of the manuscript."If this statement is not correct you must amend it as needed.Please include this amended Role of Funder statement in your cover letter; we will change the online submission form on your behalf.

Dear journal editor, hello. The sponsor plays the role of a funding provider in this study, and the source of the funding project is WANG Meng, the fourth author of this article. The author has made a statement in the cover letter.

2.When completing the data availability statement of the submission form, you indicated that you will make your data available on acceptance. We strongly recommend all authors decide on a data sharing plan before acceptance, as the process can be lengthy and hold up publication timelines. Please note that, though access restrictions are acceptable now, your entire data will need to be made freely accessible if your manuscript is accepted for publication. This policy applies to all data except where public deposition would breach compliance with the protocol approved by your research ethics board. If you are unable to adhere to our open data policy, please kindly revise your statement to explain your reasoning and we will seek the editor's input on an exemption. Please be assured that, once you have provided your new statement, the assessment of your exemption will not hold up the peer review process.

The author agrees to share research resources, including the research model code and experimental dataset, which have been sent to the journal as attachments. The dataset is too large and requires journals to provide a method of sending experimental data to the journal.

二. Comment from Reviewer # 1:

1.ABSTRACT: this needs to be re-written, summarize the methodology information and conclusion statements.

We sincerely appreciate the valuable suggestions from the reviewers. After thorough discussion among the authors, we have revised and refined the Abstract as follows:

Accurately recognizing rice seed varieties poses significant challenges due to their diverse morphological characteristics and complex classification requirements. Traditional image recognition methods often struggle with both accuracy and efficiency in this context. To address these limitations, this study proposes the Deep Space and Channel Residual Network with Double Attention Mechanism (RSCD-Net) to enhance the recognition accuracy of 36 rice seed varieties. The core innovation of RSCD-Net is the introduction of the Space and Channel Feature Extraction Residual Block (SCR-Block), which improves inter-class differentiation while minimizing redundant features, thereby optimizing computational efficiency. The RSCD-Net architecture consists of 16 layers of SCR-Blocks, structured into four convolutional stages with 3, 4, 6, and 3 units, respectively. Additionally, a Double Attention Mechanism (A2Net) is incorporated to enhance the network’s global receptive field, improving its capacity to distinguish subtle variations among seed types. Experimental results on a self-collected dataset demonstrate that RSCD-Net achieves an average accuracy of 81.94%, surpassing the baseline model by 4.16%. Compared with state-of-the-art models such as InceptionResNetV2, ConvNeXt, MobileNetV3, and Swin Transformer, RSCD Net has improved by 1.17%, 3%, 24.72%, and 13.22%, respectively, showcasing its superior performance. These findings confirm that RSCD-Net provides an effective and efficient solution for rice seed classification, offering a promising reference for addressing similar fine-grained recognition challenges in agricultural applications.

2. A few of the grammatical mistakes including punctuation etc. are found.

We sincerely appreciate the reviewers’ valuable suggestions. After discussion among the authors, the overall grammar of the paper has been polished and revised.

3. A good article may have keywords.

We sincerely appreciate the reviewers’ valuable suggestions. The papers in PLoS ONE journal and the manuscript format provided by the journal do not provide keywords. This article does not provide keywords based on journal requirements. If the journal editorial department needs keywords, this article can add them.

4. The introduction section may have contribution and objectives of the study key folded into it. The last paragraph of the introduction section may describe the layout of the of the rest of the draft.

We sincerely appreciate the reviewers’ valuable suggestions. After discussion among the authors, the last paragraph of the introduction has been revised as follows: merge the second to last paragraph with the first to last paragraph, and combine the contributions and objectives of the research together; Add a diagram describing the overall structure and arrangement of the paper after the break.

Research on the application of deep learning in rice seed classification and recognition remains limited, with few studies examining more than 15 rice varieties. In our study, we developed a dataset that encompasses 36 distinct rice seed varieties. Because rice classification belongs to fine-grained classification[19], and the characteristics between these varieties are often very similar, there may also be significant differences within the same variety, which adds complexity to the classification and recognition process. The outcomes from experiments using deep learning networks, such as ResNet, ConvNext, and InceptionResNetV2, as referenced in existing rice classification studies, have proven unsatisfactory when applied to our dataset. Additionally, while research that integrates hyperspectral imaging and 3D point cloud data may yield improved results, it also incurs higher costs. Moreover, this paper addresses another crucial challenge: many current studies depend on large quantities of training samples to train their networks. In contrast, our study seeks to effectively classify 36 different rice seed varieties using a more limited training set of only 240 samples per variety. Employing deep learning for the classification of crop seeds presents an efficient and straightforward approach to seed variety classification tasks. However, the exploration of deep learning applications specifically for rice variety classification remains underdeveloped. This study seeks to enhance the recognition accuracy of 36 rice seed varieties by leveraging deep learning technology and introducing a model titled the Deep Space and Channel Residual Network Combined with Double Attention Mechanism (RSCD-Net). The study commences with the design of a Space and channel Feature Extraction Residual Block (SCR-Block), which aims to amplify inter-class distinctions while minimizing redundancy in feature information, thereby streamlining model computation. The architecture comprises 16 residual modules, supplemented by batch normalization (BN) layers, as well as both max and average pooling layers. These residual modules are organized into a comprehensive convolutional structure featuring configurations of 3, 4, 6, and 3 to optimize feature extraction. By integrating a double attention mechanism recognized as Double Attention Networks (A2Net), the model's global receptive field is significantly enhanced, leading to improved recognition capabilities for various seed varieties. In conclusion, this study presents a rice seed classification method grounded in deep feature fusion. This innovative approach aims to tackle the high costs, time demands, and elevated misjudgment rates typically associated with traditional classification methods, providing a convenient and effective solution. The anticipated application of this method is expected to bolster the accuracy and efficiency of rice seed classification, thus fostering advancements in rice research and agricultural production. In conclusion, this study proposes a deep learning-based method for rice seed classification. This innovative approach addresses the challenges commonly associated with traditional classification methods, such as high costs, time consumption, and high misclassification rates, providing a more convenient and effective solution. Furthermore, it offers a superior framework for multi-category and fine-grained image recognition. The anticipated applications of this method are expected to enhance the accuracy and efficiency of rice seed classification, thereby contributing to advancements in rice research and agricultural production. The structure of the article is arranged as shown in Fig 1.

Fig.1 The structure of the article is arranged

5. JPG format at a resolution of 1280×720 for whole of the dataset processing seems a complex task, but authors mentioned that 224x224 image size is input?

We sincerely appreciate the reviewers’ valuable suggestions. The design of RSCD-Net is based on an input image size of 224×224 pixels. When processing images with a resolution of 1280×720, image scaling or cropping techniques are typically employed to adapt to the model’s input requirements. A common approach is to resize the image to 224×224 using resampling algorithms, which effectively reduce computational complexity while preserving essential image information. This preprocessing step ensures that the resized image meets the model’s specifications, allowing the network to efficiently extract relevant features.

The 224×224 input size is a standard configuration of the RSCD-Net model, carefully designed to balance information retention with computational efficiency. This dimension not only accelerates training speed and reduces memory consumption but also ensures that the network maintains robust visual feature learning capabilities at lower computational costs. Therefore, despite the high resolution of the original image, the model can still perform tasks such as image classification with high efficiency and accuracy through this adaptive resizing process.

For better understanding by readers, the first paragraph of the Experimental setup section in the Results chapter of the original text has been modified as follows:

We conducted experiments with 36 different varieties of rice seeds, utilizing a total of 14040 images. To thoroughly train and validate the network model, 12240 images were employed, consisting of 10800 images for the training set and 1440 images for the validation set. To maintain fairness and comparability in the experimental results, all network models were configured and executed under the same parameter settings, as detailed in Table 1. This methodology ensures an accurate assessment of each network’s performance, thereby enhancing the scientific validity and reliability of the experimental findings. The original size of the images in the dataset is 1280x720. In order to reduce computational complexity while retaining the main information of the image, the network preprocessed the image and adjusted its size to 224x224.

6. Experimental setup: capital the first letter of we->We.

We sincerely appreciate the reviewers’ valuable suggestions. Firstly, I apologize for my carelessness and have made the following revisions to this paragraph:

we We conducted experiments with 36 different varieties of rice seeds, utilizing a total of 14040 images, the specific division of the dataset is shown in Table 1. To maintain fairness and comparability in the experimental results, all network models were configured and executed under the same parameter settings, as detailed in Table 2. This methodology ensures an accurate assessment of each network’s performance, thereby enhancing the scientific validity and reliability of the experimental findings. The original size of the images in the dataset is 1280x720. In order to reduce computational complexity while retaining the main information of the image, the network preprocessed the image and adjusted its size to 224x224.

7. Learning rate, i.e., 0.0005, seems high.

We sincerely appreciate the valuable suggestions from the reviewers. The choice of a learning rate of 0.0005 is based on the following considerations:

(1)Preventing Gradient Explosion or Vanishing

A learning rate of 0.0005 is relatively small, which helps balance convergence speed and training stability. This prevents issues such as gradient explosion or vanishing, ensuring smooth parameter updates throughout the training process.

(2)Optimizing the Training of Deep Networks

RSCD-Net is a deep convolutional neural network that incorporates multiple SCR-Blocks. Training deep networks often requires a smaller learning rate, as a larger learning rate may prevent the deeper layers from receiving appropriate parameter updates, thereby hindering convergence. A smaller learning rate, such as 0.0005, allows for more stable parameter adjustments, reducing the risk of skipping over the optimal solution due to excessively large update steps.

(3)Relationship Between Learning Rate and Optimizer

When using optimizers such as Adam, a relatively small initial learning rate (e.g., 0.0005) is commonly adopted. These adaptive optimizers dynamically adjust the learning rate for each parameter, allowing for more fine-tuned updates. Consequently, a smaller initial learning rate contributes to improved training stability and convergence, ensuring that the network learns effectively.

(4)Avoiding Premature Convergence

A lower learning rate enables the model to converge gradually over an extended training period. This helps in thoroughly exploring the loss function’s optimal solution, preventing the model from becoming trapped in suboptimal local minima too early in the training process.

In summary, setting the learning rate to 0.0005 ensures stable convergence during the initial stages of training while mitigating the risks of gradient explosion or vanishing. This learning rate strikes a balance between training speed and stability, particularly for deep networks such as ResNet50. By adopting a smaller step size for updates, the network can efficiently fine-tune its weights and progressively achieve optimal performance.

8. Results are convincing.

We sincerely appreciate the reviewers' recognition of our work, which serves as a great encouragement and motivation for our research efforts.

9. Table 4, comparison is based on what? Did these models, i.e., RSCD-Net, ResNet50, ConvNext, InceptionResnetV2, and MobileNetV3 were trained on the same dataset?

We sincerely appreciate the reviewers' valuable suggestions. The networks presented in Table 4 are compared under the same evaluation metric. The experimental results shown in Table 4 represent the overall recognition accuracy of the network models for rice seeds, which is calculated as the total number of correctly identified samples divided by the total number of samples in the rice seed test set (1800).

Furthermore, RSCD-Net, ResNet50, ConvNext, InceptionResNetV2, and MobileNetV3 were trained and evaluated on the same dataset, as stated in the first paragraph of the Ablation Experiment section of the article. The or

---

## [Decision Letter · Decision Letter 1]

26 Mar 2025

Multi-class Rice Seed Recognition Based on Deep Space and Channel Residual Network Combined with Double Attention Mechanism

PONE-D-24-54192R1

Dear Dr. Changsheng,

We’re pleased to inform you that your manuscript has been judged scientifically suitable for publication and will be formally accepted for publication once it meets all outstanding technical requirements.

Kind regards,

Xiaoyong Sun

Academic Editor

PLOS ONE

sunx1@sdau.edu.cn

Additional Editor Comments (optional):

Reviewers' comments:

Reviewer's Responses to Questions

**Comments to the Author**

1. If the authors have adequately addressed your comments raised in a previous round of review and you feel that this manuscript is now acceptable for publication, you may indicate that here to bypass the “Comments to the Author” section, enter your conflict of interest statement in the “Confidential to Editor” section, and submit your "Accept" recommendation.

Reviewer #1: All comments have been addressed

Reviewer #2: All comments have been addressed

2. Is the manuscript technically sound, and do the data support the conclusions?

Reviewer #1: Yes

Reviewer #2: Yes

3. Has the statistical analysis been performed appropriately and rigorously? 

Reviewer #1: Yes

Reviewer #2: Yes

4. Have the authors made all data underlying the findings in their manuscript fully available?

Reviewer #1: Yes

Reviewer #2: Yes

5. Is the manuscript presented in an intelligible fashion and written in standard English?

Reviewer #1: Yes

Reviewer #2: Yes

6. Review Comments to the Author

Reviewer #1: The authors addressed all of my comments in well manner. Therefore, acceptance is suggested in its current form.

Reviewer #2: The authors have successfully addressed all the comments and suggestions provided in the initial review. The revisions made are thorough, demonstrating a clear understanding of the feedback, and have significantly improved the clarity, structure, and overall quality of the manuscript. The adjustments to the methodology have strengthened the robustness of the study, and I appreciate the authors’ diligent efforts to incorporate the suggestions. I am satisfied with the revisions and pleased to accept the paper for publication. This work provides valuable insights and will contribute positively to the field, and I look forward to seeing future developments and related research from the authors.

7. PLOS authors have the option to publish the peer review history of their article (what does this mean? ). If published, this will include your full peer review and any attached files.

**Do you want your identity to be public for this peer review?** For information about this choice, including consent withdrawal, please see our Privacy Policy .

Reviewer #1: **Yes: ** Ghulam Gilanie

Reviewer #2: No

---

## [Editor Report · Acceptance letter]

PONE-D-24-54192R1

PLOS ONE

Dear Dr. Zhang,

I'm pleased to inform you that your manuscript has been deemed suitable for publication in PLOS ONE. Congratulations! Your manuscript is now being handed over to our production team.

Kind regards,

on behalf of

Dr. Xiaoyong Sun

Academic Editor

PLOS ONE